# Neutrophil extracellular traps promote immunopathogenesis of virus-induced COPD exacerbations

Orestis Katsoulis[1], Marie Toussaint[2], Millie M. Jackson[1], Patrick Mallia[2], Joseph Footitt[2], Kyle T. Mincham[2], Garance F. M. Meyer[2], Tata Kebadze[2], Amy Gilmour[3], Merete Long[3], Andrew D. Aswani[4], Robert J. Snelgrove[2], Sebastian L. Johnston [ID][2], James D. Chalmers[3] & Aran Singanayagam [ID][1,2] ✉

Respiratory viruses are a major trigger of exacerbations in chronic obstructive pulmonary disease (COPD). Airway neutrophilia is a hallmark feature of stable and exacerbated COPD but roles played by neutrophil extracellular traps (NETS) in driving disease pathogenesis are unclear. Here, using human studies of experimentally-induced and naturally-occurring exacerbations we identify that rhinovirus infection induces airway NET formation which is amplified in COPD and correlates with magnitude of inflammation and clinical exacerbation severity. We show that inhibiting NETosis protects mice from immunopathology in a model of virus-exacerbated COPD. NETs drive inflammation during exacerbations through release of double stranded DNA (dsDNA) and administration of DNAse in mice has similar protective effects. Thus, NETosis, through release of dsDNA, has a functional role in the pathogenesis of COPD exacerbations. These studies open up the potential for therapeutic targeting of NETs or dsDNA as a strategy for treating virus-exacerbated COPD.

COPD is a disease that is punctuated by the occurrence of acute exacerbations, most commonly triggered by respiratory virus infections[1]. Exacerbations are a major cause of morbidity and mortality and are responsible for a significant medical and socioeconomic burden. Human experimental challenge models have demonstrated a causal role for rhinoviruses in precipitating acute exacerbations with greater airway inflammation, symptoms, and acute lung function decline observed in patients with COPD compared to control participants in these studies[2,3]. However, the pathobiological mechanisms that drive acute exacerbations remain poorly understood; a greater understanding could facilitate the development of new effective therapies.

Airway neutrophilia is a hallmark feature of COPD[4,5] and is further increased during virus induced-exacerbations[2,6]. Neutrophil recruitment subsequently leads to the formation of neutrophil extracellular traps (NET), web-like scaffolds containing host extracellular double stranded DNA (dsDNA) in conjunction with histones, myeloperoxidase, and neutrophil elastase. NETs are recognised to have beneficial and detrimental roles depending on the specific disease context. Antiviral properties of NETs have been reported against poxviruses[7] although this has not been shown in the context of respiratory viruses such as influenza[8]. Conversely, sustained NET formation is recognised to drive immunopathology. This has been most widely demonstrated in COVID-19 where systemic and airway NET accumulation induces hyperinflammation during the acute and chronic disease phases[9,10]. NET formation leads to release of dsDNA into the extracellular space and evidence suggests that during viral infection, host dsDNA is released extracellularly as a damage-associated molecular pattern (DAMP) and can be recognised by and modulate the immune system to promote host defence[11–14]. We have previously reported that rhinovirus

---

[1]Department of Infectious Disease, Centre for Bacterial Resistance Biology, Imperial College London, London, UK. [2]National Heart and Lung Institute, Imperial College London, London, UK. [3]Division of Molecular and Clinical Medicine, University of Dundee, Ninewells Hospital and Medical School, Dundee, UK. [4]Department of Intensive Care Medicine, Guy's and St Thomas' NHS Foundation Trust, London SE1 7EH, UK. ✉e-mail: a.singanayagam@imperial.ac.uk

(RV) infection induces NET formation during asthma exacerbations leading to dsDNA release which amplifies inflammation and enhances exacerbation severity[15]. A similarly important role of dsDNA may be expected to occur in COPD, where the neutrophil plays an even more prominent role in exacerbations by promoting airway inflammation and mucus hypersecretion, and thus potentially increasing exacerbation severity.

Excessive NET production has been demonstrated in stable and exacerbated COPD and correlates with inflammation, microbial dysbiosis, airflow limitation, and frequent exacerbations[16–18]. Pharmacological inhibition of NETs in a mouse model of cigarette smoke extract (CSE)-induced COPD attenuated pathological features including extent of histological emphysema, airway leukocyte infiltration, and mucus hypersecretion[19]. Collectively, these data indicate that NETs are a major component of the COPD airway milieu and may have pathological consequences. However, their role as a mechanistic driver of exacerbation pathogenesis in COPD has not been elucidated.

Here, we show that RV infection induces formation of NETs in the airways, which is amplified in COPD relative to health and correlates with immunopathology and exacerbation severity. Pharmacological inhibition of NETs in a mouse model of virus-exacerbated COPD reduces inflammation and ameliorates lung function. Similar effects

were induced through administration of DNAse in virus-exacerbated mice, indicating that NETs may induce downstream immunopathology through release of extracellular dsDNA. Thus, NETosis, by releasing dsDNA, plays a central functional role in COPD exacerbation pathogenesis.

## Results

### Experimental rhinovirus infection induces airway NET formation in COPD patients

We have previously reported that RV infection in COPD is associated with enhanced airway inflammation with increased total cells, neutrophils, lymphocytes, and a range of inflammatory chemokines and cytokines all increased in patients with COPD versus control participants[2,3]. Initially, to investigate whether NETosis is induced by viral infection and similarly exacerbated in COPD, we quantified surrogate NET markers by ELISA within a study of human rhinovirus challenge in COPD and healthy control participants (Fig. 1A). Baseline characteristics of included subjects are shown in Table 1. Sputum concentrations of DNA-elastase complexes (a well-established marker of NETosis[16]) were induced from baseline at day 9 in COPD participants but not healthy control smokers or non-smokers (Fig. 1B). Day 9 concentrations of sputum DNA-elastase were increased in COPD versus

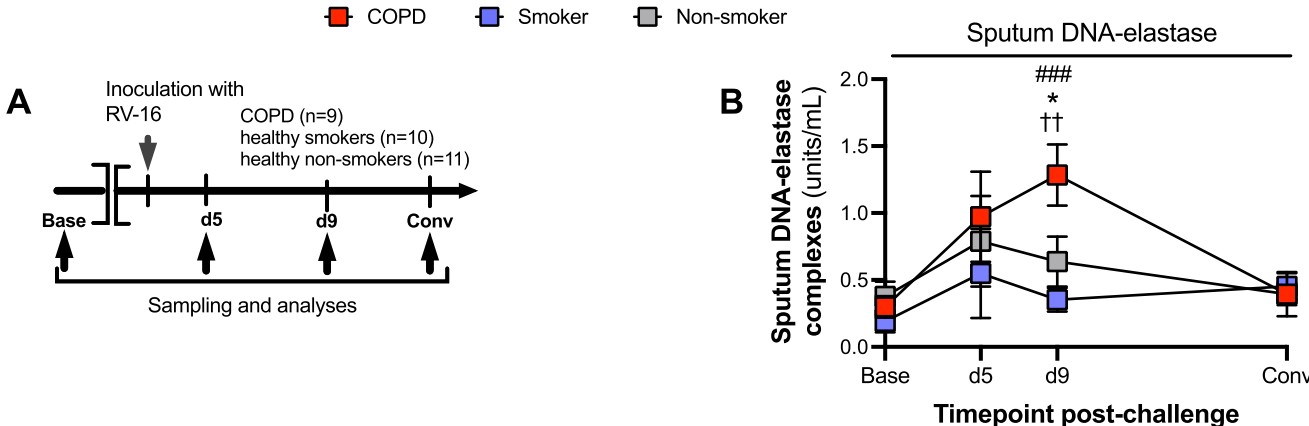

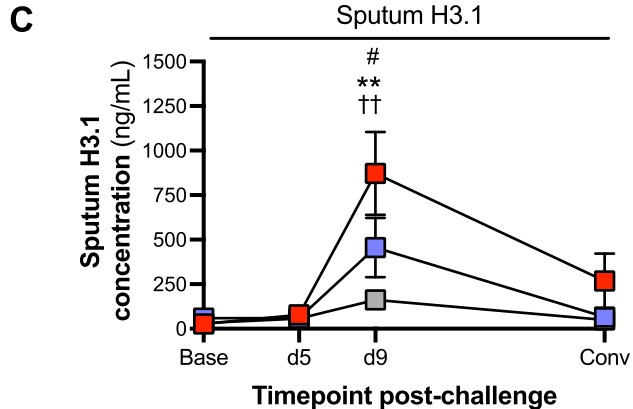

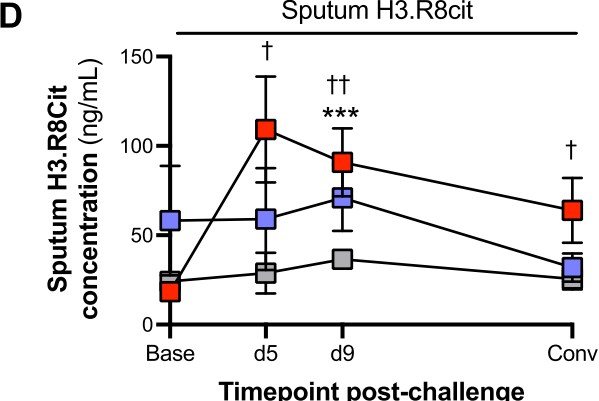

**Fig. 1 | NETosis is increased during human RV infection in COPD. A** Experiment schematic. 9 participants with chronic obstructive pulmonary disease (COPD), 10 healthy smokers and 11 healthy non-smokers underwent sampling at baseline, day 5, day 9 and convalescence during experimental RV-A16 challenge. **B** Sputum DNA/elastase complexes (**C**) total intact (H3.1) and (**D**) citrullinated (H3.R8) nucleosomes were quantified by ELISA. †$†P < 0.01$ COPD group day 9 vs baseline; †$P < 0.05$ COPD group day 5 or 42 vs baseline; ***$P < 0.001$ COPD vs. non-smokers **$P < 0.01$ COPD

vs. non-smokers. *$P < 0.05$ COPD vs. non-smokers. ###$P = 0.001$ COPD vs smokers. #$P < 0.05$ COPD vs smokers. Data are presented as mean values ± SEM. Data analysed by two-tailed Wilcoxon matched pairs, signed-rank test to compare baseline to post-infection or two-tailed Mann–Whitney test to compare between experimental groups. $n = 9$ in each group in (**B**) due to sample availability. Source data are provided as a Source Data file.

**Table 1 | Baseline characteristics of participants included in human viral challenge study**

| | COPD (n = 9) | Healthy Smokers (n = 10) | Healthy Non-smokers (n = 11) |
|---|---|---|---|
| Age (mean (SD)) | 60.4 (3.2) | 52.5 (2.2) | 62.2 (1.6) |
| Sex (% female) | 33.3% | 60.0% | 63.6% |
| Smoking history (pack yrs, mean(SD)) | 39.4 (3.3) | 32.1 (3.0) | n/a |
| FEV1 (% predicted, mean(SD)) | 68.1 (1.6) | 96.6 (3.3) | 102.2 (3.3) |

healthy smokers and non-smokers. NETosis involves hypercitrullination of nucleosomal histone proteins, including H3, which can be quantified as remnant of NET formation[9]. We therefore measured total intact (H3.1) and citrullinated (H3R8cit) nucleosomes by ELISA at selected timepoints encompassing the early (day 5), late (day 9) and recovery (day 42) phases of RV infection. We found that total H3.1 is similarly induced at day 9 from baseline in the COPD group but not in the control groups (Fig. 1C). Total H3R8 citrullinated nucleosomes were induced at days 5, 9 and 42 from baseline in the COPD group, but not in the control groups (Fig. 1D). At day 9 post-infection, concentrations of H3.1 were increased in COPD versus healthy smokers and non-smokers and concentrations of H3R8cit were increased in COPD versus non-smokers (Fig. 1C, D). We additionally assessed measurements of sputum DNA-elastase, H3.1, and H3R8cit after adjustment for total sputum neutrophil counts which showed no significant differences between the three study groups, indicating that increased NETosis in COPD is likely due to increased neutrophil recruitment during experimental viral challenge (Supplementary Fig. 1). We quantified sputum myeloperoxidase (a further marker of neutrophil activation) in sputum samples from this model but this marker was not significantly induced in any of the groups (Supplementary Fig. 2).

Collectively, these data indicated that NET formation is upregulated during virus-induced COPD exacerbation, findings that are consistent with our prior observations in this model that sputum neutrophil elastase (a major constituent of NETs) is also augmented in COPD compared to control donors[3].

## NETs correlate with immunological and clinical exacerbation severity during experimentally induced exacerbations

Having established that RV infection induces airway NET formation in COPD patients, we next assessed whether total H3.1 and markers of NETosis DNA-elastase and H3.R8Cit correlated with inflammation and exacerbation severity. All three parameters significantly correlated with cellular airway inflammation (total cell count and neutrophils) and concentrations of neutrophil elastase (Fig. 2A–C). H3.1 and H3R8cit correlated with sputum concentrations of a range of inflammatory cytokines (IL-1β, IL-4, IL-5, IL-6, IP-10, and TNF) and the mucin glycoprotein MUC5AC. Sputum DNA-elastase complexes showed correlations with IL-1β, IL-8, and IP-10 (Fig. 2A–C). All three parameters also correlated with virus loads (Fig. 2D–F). Although sputum DNA-elastase did not correlate with lower respiratory tract symptom scores (Fig. 2G) significant correlations were observed for H3.1 and H3.R8Cit (Fig. 2H, I). There were also non-significant trends towards a negative correlations with acute decline in peak expiratory flow rate (in COPD participants alone) (Fig. 2J–L).

## NETs are elevated during naturally occurring virus-associated COPD exacerbations

We further investigated NETosis during 'real-world' naturally occurring virus-associated exacerbations of COPD in a cohort (n = 18) that had been prospectively monitored in the community (Fig. 3A). Baseline

characteristics of the study cohort are shown in Table 2. Sputum DNA/elastase complexes were significantly induced at exacerbation onset from stable-state with resolution down to steady-state levels by two weeks (Fig. 3B). Sputum H3.1 showed trends towards increases at exacerbation onset and 2 weeks that failed to reach statistical significance (Fig. 3C). H3R8 was significantly elevated at two weeks post-onset with a non-significant trend towards increase (P = 0.058, Wilcoxon matched-pairs, signed-rank test) observed at onset (Fig. 3D). As in the human challenge study, we again assessed measurements of sputum DNA-elastase, H3.1, and H3R8cit after adjustment for total sputum neutrophil counts (in a subset where these data were available) (Supplementary Fig. 3).

Sub-stratification of participants in this study into mild (GOLD stage I–II) and severe (GOLD stage III–IV) spirometric disease indicated that sputum DNA-elastase complexes were more prominently induced in individuals with severe disease with no clear differences for sputum H3.1 and trends towards the opposite effect observed for sputum H3R8Cit (Supplementary Fig. 4). Consistent with findings in the experimental challenge exacerbation model, sputum neutrophil elastase was induced from stable-state to exacerbation, an effect not observed for sputum MPO (Supplementary Fig. 5).

These data corroborated our findings in experimentally-induced exacerbations by showing that increased NETosis also occurs during naturally occurring virus-induced exacerbations.

## A mouse model of rhinovirus infection in elastase-induced COPD recapitulates human exacerbation neutrophil and NET biology

Having observed that NETs are increased in experimentally-induced and naturally-occurring human COPD exacerbations, we next sought to investigate causal roles played in driving exacerbation pathogenesis using functional experiments in mouse models. We have previously reported that porcine pancreatic elastase administration combined with rhinovirus infection into the airways of mice (Fig. 4A) recapitulates the key features of human COPD exacerbation including augmented airway (BAL) neutrophilic inflammation and enhanced induction of the neutrophil products myeloperoxidase and elastase[20,21]. We further characterised this model to show that rhinovirus infection exacerbates lung tissue recruitment of total neutrophils and activated CD63+ and CD64+ neutrophils in elastase-treated mice (Fig. 4B–D). Previous studies have demonstrated that lung neutrophils expressing CXCR4 and Lamp-1 display a heightened propensity to release NETS[22]. Accordingly, the percentage of neutrophils that were CXCR4high and Lamp-1high were increased in the lungs of elastase-treated, RV infected mice versus PBS-treated RV-infected control mice (Fig. 4E, F). In keeping with our analyses in human disease models (Figs. 1, 3), RV infection also led to augmented BAL concentrations of total and citrullinated histones H3.1 and H3.R8cit in elastase-treated mice compared to controls (Fig. 4G, H). Analysis of lactate dehydrogenase (LDH) concentrations in BAL indicated a trend towards lower concentrations in elastase-treated RV infected mice versus PBS-treated RV infected control mice (Fig. 4I), suggesting that augmented NETosis observed in elastase-treated mice was not due to increased cellular damage during viral infection.

Collectively, these data confirmed that rhinovirus infection in elastase-treated mice recapitulates similar features of NETosis as observed in our human exacerbation studies and provided validation for use of this model to further study functional roles of NETs.

## Pharmacological blockade of neutrophil elastase reduces immunopathology and severity of virus-exacerbated COPD

We next carried out manipulation experiments in the elastase mouse model of exacerbated COPD to investigate whether targeting of neutrophil elastase (a major constituent of NETs) would have effects upon viral exacerbation pathogenesis.

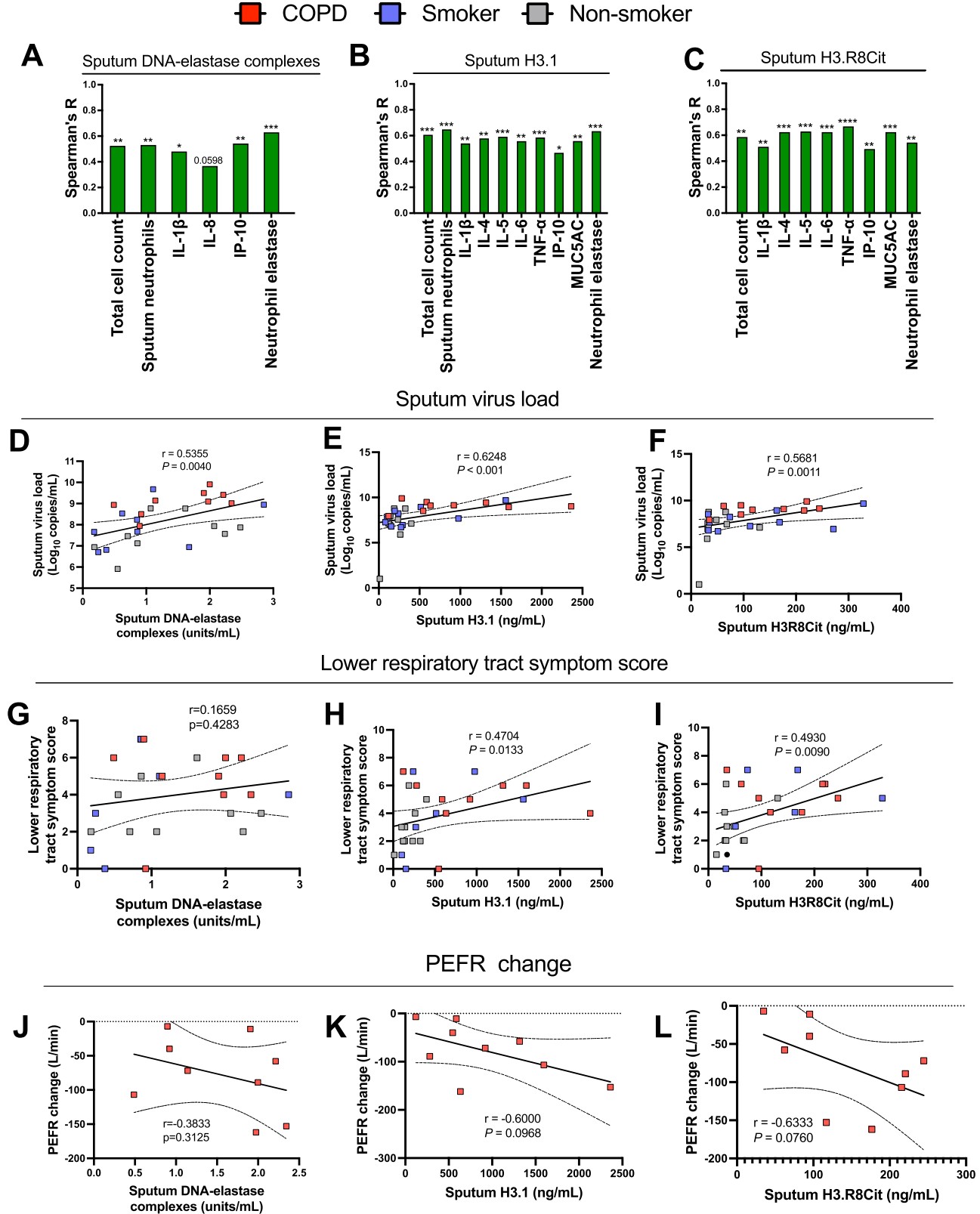

**Fig. 2 | Airway NETs correlate with immunological and clinical COPD exacerbation severity.** Correlation of sputum DNA-elastase complexes, total intact (H3.1) and citrullinated (H3.R8) nucleosomes with (**A**–**C**) cellular airway inflammation, cytokines, MUC5AC and neutrophil elastase. **D**–**F** sputum virus loads. **G**–**I** lower respiratory tract symptoms scores. **J**–**L** PEFR change. Correlation analysis used was non-parametric (Spearman's correlation) performed on COPD ($n = 9$), healthy smoker ($n = 10$) and healthy non-smoker ($n = 11$) participants pooled into a single group ***$P < 0.001$, **$P < 0.01$, *$P < 0.05$. Source data are provided as a Source Data file.

**Table 2 | Baseline characteristics of participants included in naturally occurring exacerbation study**

| Characteristic | |
|---|---|
| Age | 68.2 (10.3) |
| Sex (% female) | 38.9% |
| Smoking history (pack yrs, mean (SD)) | 47.5 (40) |
| FEV 1 (%predicted, mean (SD)) | 65 (20.7) |
| Inhaled corticosteroid use | 38.9% |
| Long acting bronchodilator use | 50% |
| Antibiotics initiated for exacerbation | 44.4% |
| Prednisolone initiated for exacerbation | 16.7% |

Mice with elastase-induced COPD like changes were treated intraperitoneally with a specific inhibitor of neutrophil elastase (GW311616A) prior to infection with RV-A1B (Fig. 5A) using a dosing schedule we have previously optimised and reported[15]. Neutrophil elastase inhibitor therapy attenuated airway concentrations of total and citrullinated nucleosomes H3.1 and H3R8cit at 24 h post-infection (Fig. 5B). Neutrophil elastase inhibitor therapy also concurrently reduced RV induction of histological inflammation scores (Fig. 5C), cellular airway inflammation (BAL total cells, neutrophils and macrophages with no effect observed on lymphocytes, Fig. 5D) and cytokine expression including the chemokines CXCL10/IP-10 and CCL5/RANTES and the pro-inflammatory mediators TNF, IL-1β and IL-6 (Fig. 5E, F) at 24 h post-infection. Neutrophil elastase inhibitor therapy led to a non-statistically significant (P = 0.11, two-tailed Mann–Whitney U test) reduction in RV induction of BAL MUC5AC concentrations at day 4 post-infection (Fig. 5G). Neutrophil elastase inhibitor therapy also augmented RV-induction of the antimicrobial peptide secretory leucocyte protease inhibitor (SLPI) (Fig. 5H), a protein we have previously shown to drive secondary bacterial infection during virus exacerbated COPD[2,23]. Neutrophil elastase inhibitor therapy abrogated RV-induced airway hyper-responsiveness (AHR) to methacholine challenge in elastase-treated mice at day 1 post-infection (Fig. 5I). There was no evidence of neutrophil elastase inhibition leading to downstream activation of pathways promoting neutrophil recruitment or persistence, as neutrophil counts at day 4 were not increased and showed a trend towards remaining reduced (Supplementary Fig 6). Despite altering early airway pro-inflammatory responses, neutrophil elastase inhibitor therapy had no effect on anti-viral immune responses (BAL concentrations of IFN-α, Supplementary Fig. 7A) or virus replication (Supplementary Fig. 7B) in this model.

These data demonstrated that targeting neutrophil elastase in a pre-clinical COPD exacerbation model reduces NET formation and attenuates virus-induced immunopathology and exacerbation severity. Coupled with our findings in human viral COPD exacerbations, this indicates a functional role for neutrophil elastase/NETosis in driving exacerbation pathogenesis in COPD.

**Host dsDNA is released during experimental rhinovirus infection, correlates with virus loads and is accentuated in COPD**

We next hypothesised that NETs may induce their deleterious effects on immunopathology and exacerbation severity through release of

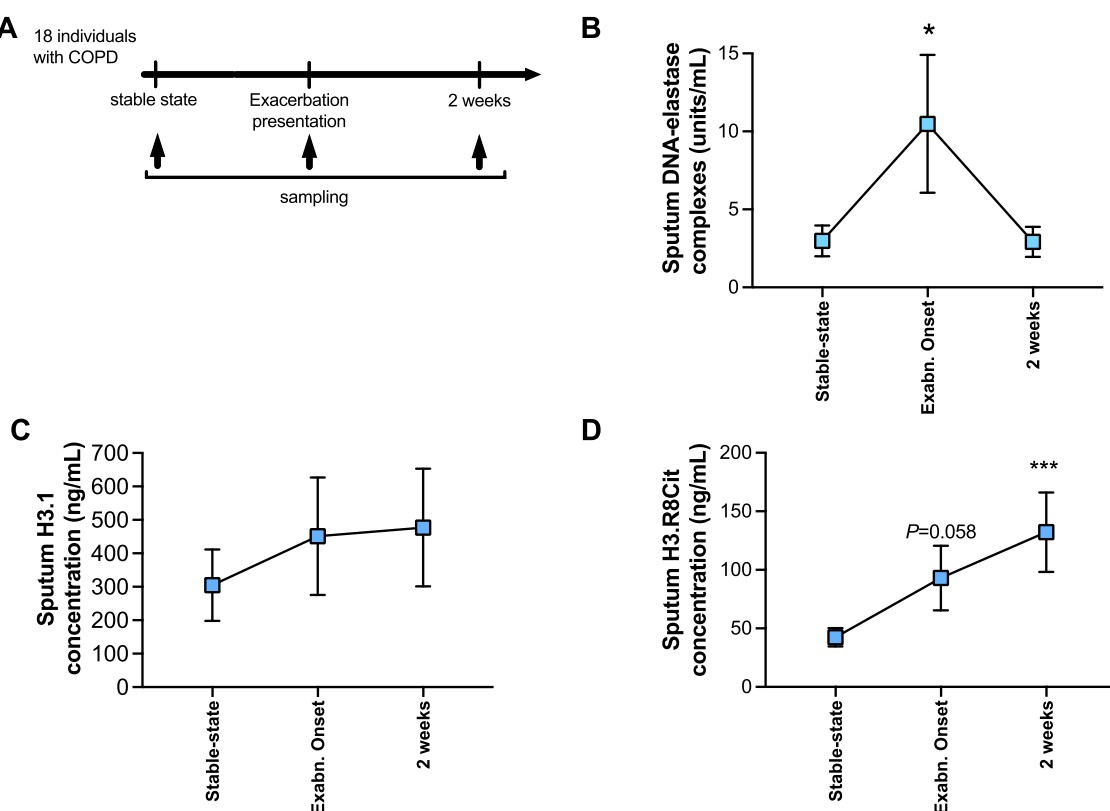

**Fig. 3 | Airway NETs are increased during naturally occurring virus associated exacerbations. A** Experiment schematic. 18 individuals with chronic obstructive pulmonary disease (COPD) were monitored prospectively. Sputum samples were taken during stable state (baseline), at presentation with an exacerbation associated with positive virus detection, and 2 weeks after exacerbation presentation. Sputum concentrations of (**B**) DNA-elastase complexes, (**C**) total intact (H3.1) and (**D**) citrullinated (H3.R8) nucleosomes were measured at stable-state and following virus-induced exacerbation. Data are presented as mean values ± SEM. *P < 0.05 ***P < 0.001,. Data analysed by two-tailed Wilcoxon matched pairs, signed-rank test to compare steady-state with exacerbation onset or 2 weeks after exacerbation. Source data are provided as a Source Data file.

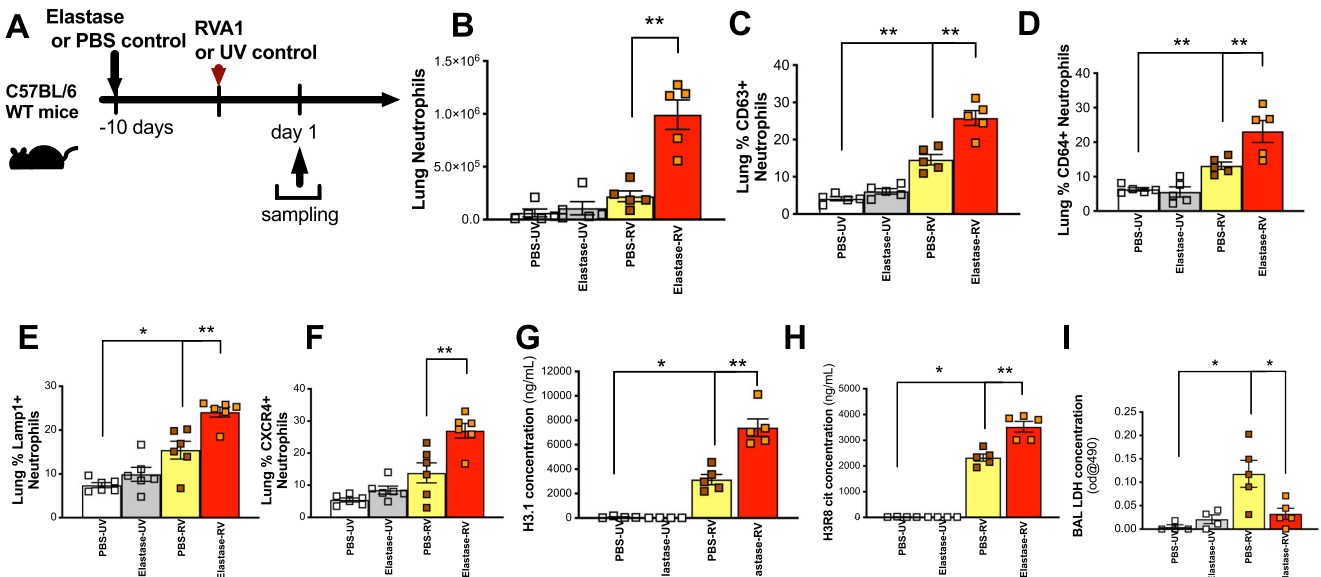

**Fig. 4 | Elastase treatment combined with rhinovirus infection models exacerbated neutrophils and NETs in mice. A** Experimental schematic. Mice were treated with intranasal porcine pancreatic elastase or PBS control. 10 days later, mice were inoculated with RVA1 or UV-inactivated virus control. **B** Total neutrophil numbers in lung tissue (**C**) % CD63+ (**D**) % CD64+ (**E**) Lamp-1 and (**F**) CXCR4 expressing neutrophils were characterised by flow cytometry.

Bronchoalveolar lavage (BAL) concentrations of (**G**) total intact (H3.1) (**H**) citrullinated (H3.R8) nucleosomes and (**I**) lactate dehydrogenase (LDH) were quantified by ELISA. $n = 4–6$ mice/group, representative of at least two independent experiments. Data presented as mean ± SEM. Data analysed by two-tailed Mann–Whitney U test. $*P < 0.05$, $**P < 0.01$,. Source data are provided as a Source Data file.

extracellular DNA sequestered during viral infections. We therefore initially quantified dsDNA concentrations in sputum samples from the same human experimental viral challenge studies (Fig. 6A). Baseline demographic characteristics for all participants are again shown in Table 1. RVs are RNA viruses with no dsDNA intermediates[24] and therefore, dsDNA detected in airway samples following RV infection is solely host-derived. RV infection significantly increased dsDNA levels from baseline in sputum samples from patients with COPD on day 9, 12 and 15 post-infection and in smokers on day 9 post-infection with no significant induction observed in healthy non-smokers (Fig. 6B). Significantly increased sputum DNA levels were observed in patients with COPD versus healthy non-smokers on day 9 and 15 post-infection and versus smokers at day 9, 12 and 15 post-infection (Fig. 6B). As expected, dsDNA concentrations strongly correlated with concentrations of total and citrullinated nucleosomes H3.1 and H3R8cit (Fig. 6C). These data suggested that virus-induced airway host dsDNA may be released from NETs and accentuated in COPD participants during acute infection.

## Host dsDNA correlates with immunopathology and clinical exacerbation severity

We next assessed whether levels of host dsDNA during exacerbation correlated with virus burden and virus-induced airway inflammation. In all participants combined, dsDNA levels correlated with sputum virus loads (Fig. 6D), sputum total cells, neutrophil numbers (Fig. 6E) and sputum levels of the inflammatory cytokines IL-1β, IL-4, IL-5, IL-8, IL-12 CXCL10/IP-10, MDC, TARC, eotaxin 1 and GM-CSF (Fig. 6F). dsDNA additionally correlated with sputum MUC5AC (Fig. 6G), upper and lower respiratory symptom scores (Fig. 6H).

## Host DNA is released during naturally occurring exacerbations

We further studied dsDNA concentrations in our cohort of naturally occurring virus-associated exacerbations. Baseline characteristics are again shown in Table 2. Sputum dsDNA concentrations increased at exacerbation onset compared to stable state with persistently increased levels but some evidence of resolution observed by 2 weeks post-onset (Supplementary Fig. 8). These data corroborated our

findings in experimental infections and indicated that host DNA increases during virally driven COPD exacerbations.

## Therapeutic inhibition of DNA ameliorates immunopathology and severity in a mouse model of virus-exacerbated COPD

Having observed that extracellular DNA correlates with similar clinical and immune parameters to that observed for NETs in our human model, we next interrogated its functional role in driving RV immunopathology within the elastase mouse model through administration of the DNA-hydrolysing enzyme DNAse I. Elastase-treated mice were administered a combination of intraperitoneal and intranasal DNAse I, using a dosing strategy we have previously optimised and reported[15], prior to infection with RVA1 (Fig. 7A). DNAse administration led to potent suppression of BAL DNA concentrations in RV-infected elastase-treated mice at day 1 and day 4 post-infection (Fig. 7B). This was associated with attenuated airway inflammatory cell recruitment (Fig. 7C) and reduced chemokine (CXCL10, CCL5, CCL2, Fig. 7D) and cytokine (IL-6, IL-1β, TNF, Fig. 7E) induction. RV-induction of MUC5AC and AHR assessed by whole body plethysmography were also significantly reduced by DNAse administration in elastase-treated mice (Fig. 7F, G). As observed with Neutrophil elastase inhibitor therapy, DNase administration had no effect on anti-viral immune responses (BAL concentrations of IFN-α Supplementary Fig. 9A) or virus replication (Supplementary Fig. 9B)in this model.

These data indicated that similar effects on RV-induced responses to those observed with neutrophil elastase inhibition could be induced by DNAse administration, supporting the mechanistic hypothesis that NETs induce downstream immunopathology through release of extracellular DNA which is therefore also a viable therapeutic target in COPD.

## Specific inhibition of NETs reduces RV-induction of extracellular DNA and airway inflammation

To further probe the link between NETs, DNA and virus-induced inflammation, we utilised intraperitoneal administration of an inhibitor of the arginine deiminase PAD (BB-Cl-amidine) within the elastase

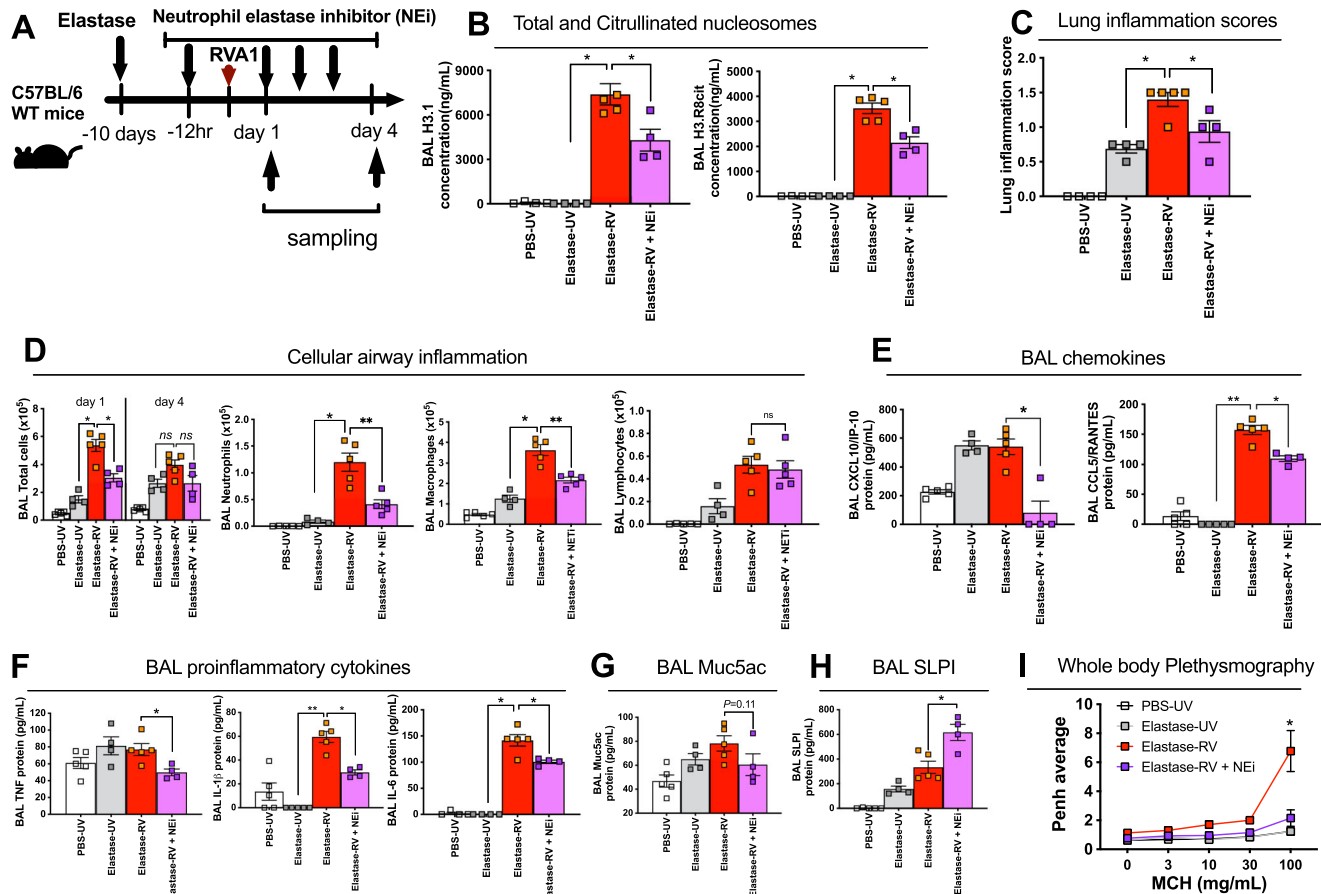

**Fig. 5 | Pharmacological neutrophil elastase inhibition reduces immuno-pathology in a mouse model of virus-exacerbated COPD. A** Experimental schematic. Mice were treated with intranasal porcine pancreatic elastase or PBS control. 10 days later, mice received i.p. injection of the Neutrophil elastase inhibitor (NEi; GW311616A) or vehicle control 12 h before RVA1 inoculation. **B** Bronchoalveolar lavage (BAL) concentrations of Total intact (H3.1) and citrullinated (H3.R8) nucleosomes were quantified by ELISA. **C** Histological lung inflammation scores. **D** BAL total cells, neutrophils, macrophages and lymphocytes were enumerated by cytospin. BAL concentrations of (**E**) chemokines CXCL10/IP-10 and CCL5/RANTES (**F**) pro-inflammatory cytokines TNF, IL-1β and IL-6, (**G**) Muc5ac and (**H**) secretory leucocyte protease inhibitor (SLPI) were quantified by ELISA. (**I**) Airway hyperresponsiveness (enhanced pause [Penh]) to methacholine challenge was measured by whole body plethysmography. $n = 4$-5 mice/group, representative of at least two independent experiments. Data presented as mean ± SEM. Data analysed by two-tailed Mann–Whitney U test. *$P < 0.05$, **$P < 0.01$,. Source data are provided as a Source Data file.

mouse model (Fig. 8A). PAD-inhibition attenuated airway concentrations of total and citrullinated nucleosomes H3.1 and H3R8cit at 24 h post-infection (~87% and 94% inhibition respectively, Fig. 8B), confirming a potent suppression of NETosis. PAD-inhibition also potently suppressed BAL DNA concentrations in RV-infected elastase-treated mice (Fig. 8C), providing direct causal evidence that virus-induced airway host dsDNA is released from NETs during acute infection.

As observed for neutrophil elastase and extracellular DNA inhibition, PAD-inhibitor treatment attenuated airway inflammatory cell recruitment (Fig. 8D), chemokine (CXCL10 and CCL5, Fig. 8E) and pro-inflammatory cytokine responses (IL-6 and IL-1β but not TNF, Fig. 8F) to RV. However, PAD inhibitor treatment did not impact upon RV-induction of MUC5AC (Fig. 8G). As observed for neutrophil elastase inhibition, there was no evidence of PAD inhibition leading to downstream activation of pathways promoting neutrophil recruitment or persistence as neutrophil counts at day 4 showed a trend towards being reduced in the PAD-treated group (Supplementary Fig. 10).

## Discussion

Through a combination of immune profiling in samples taken from human studies of viral infection in COPD coupled with functional experiments in animal models, we elucidate a mechanistic role for NETs in driving the immunopathogenesis of exacerbations.

We identify that NETosis, and consequent dsDNA released into the airways during viral infection, induces airway inflammation and mucus hypersecretion, culminating in clinical exacerbation. Accordingly, therapeutic targeting of NETs or dsDNA during virus-exacerbated disease ameliorated downstream immunopathology and airways hyperresponsiveness within a preclinical mouse model.

COPD is the third leading cause of death worldwide and a disease punctuated by the occurrence of exacerbations, most commonly triggered by rhinovirus. These episodes are responsible for a major burden of morbidity and mortality. Methods for treating exacerbations (e.g., corticosteroids, antibiotics) have remained unchanged for many years and there is an urgent need for new biological understanding to identify novel targets for intervention. Neutrophils have long been recognised to be a hallmark feature in stable COPD present in 60–80% of patients[4,5] and are further increased during exacerbations[2,6] Despite this, attempts to target neutrophilic inflammation (e.g., CXCR2 antagonists, IL-17 monoclonal antibodies) have largely failed to show efficacy in clinical trials[25–27]. Targeting neutrophils directly may also be sub-optimal due to the delicate balance that exists between reducing inflammation versus increasing the risk of infections through immunosuppression. Therefore, focus has shifted towards downstream targets including NETs and neutrophil proteases where these risks may theoretically be lesser. NETs have well

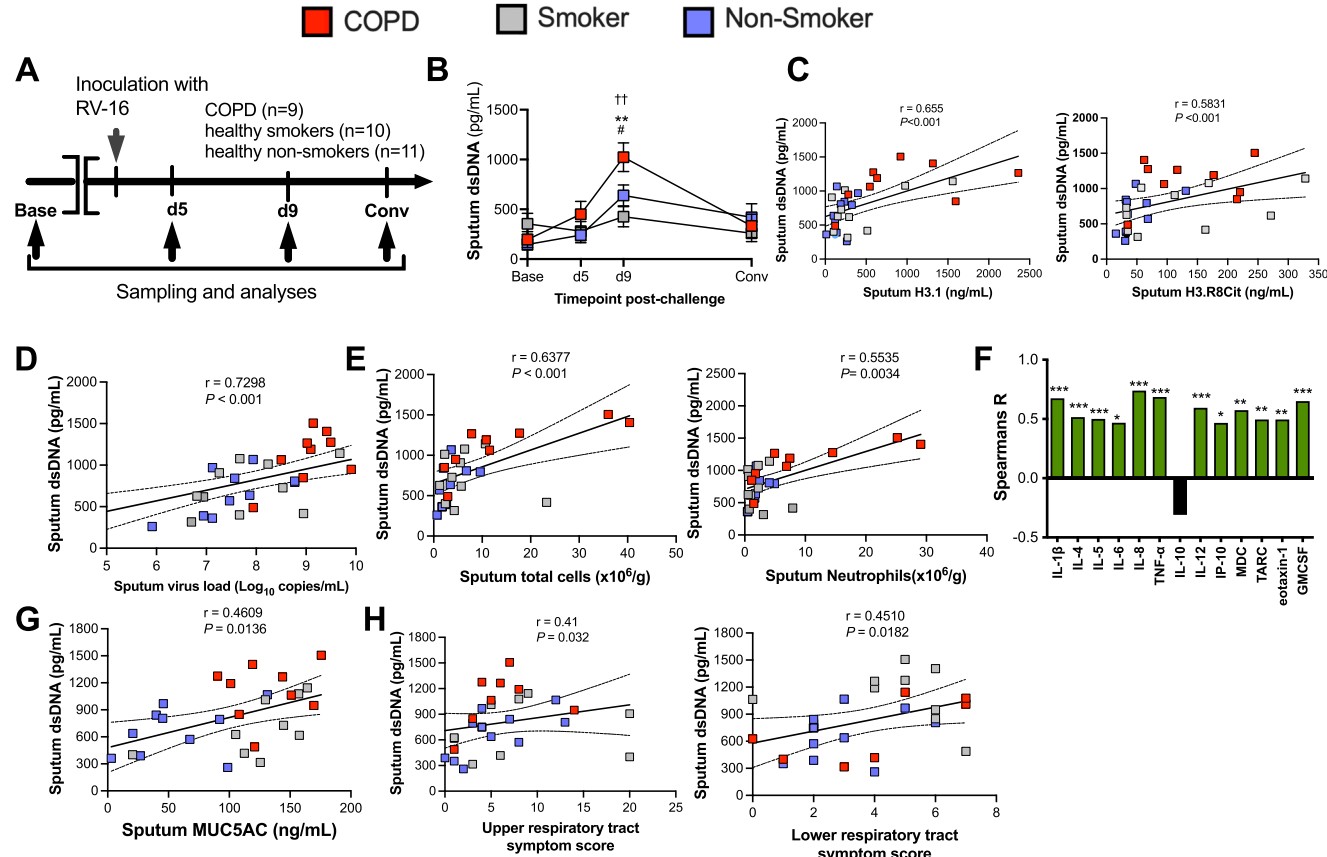

**Fig. 6 | Host dsDNA induction is increased during human RV infection in COPD and correlates with immunopathology and clinical exacerbation severity.**
**A** Experimental schematic. 9 participants with chronic obstructive pulmonary disease (COPD), 10 healthy smokers and 11 healthy non-smokers underwent sampling at baseline, day 5, day 9 and convalescence during experimental rhinovirus infection. **B** Sputum dsDNA concentrations following RV-A16 challenge in 9 COPD, 10 healthy smoker and 10 healthy non-smoker control participants. Correlation of peak sputum dsDNA concentrations with (**C**) sputum concentrations of Total intact (H3.1) and citrullinated (H3.R8) nucleosomes, (**D**) sputum Virus loads (**E**) sputum total and neutrophil cell counts (**F**) sputum cytokine concentrations and (**G**)

sputum MUC5AC concentrations. Correlation of peak sputum dsDNA with (**H**) Upper and Lower respiratory tract symptoms. In (**B**) ††P < 0.01 COPD group day 9 vs baseline. Data in (**B**) are presented as mean values ± SEM and analysed by two-tailed Wilcoxon matched pairs, signed-rank test to compare baseline to post-infection or two-tailed Mann–Whitney test to compare between experimental groups. **P < 0.01 COPD vs. non-smokers #p < 0.05 COPD vs smoker. In (**C**–**I**) correlation analysis used was non-parametric (Spearman's correlation) performed on COPD (n = 9), healthy smoker (n = 10) and healthy non-smoker (n = 11) participants pooled into a single group. Source data are provided as a Source Data file.

recognised antimicrobial effects, particularly in the context of bacterial infections[28]. However, it is increasingly recognised that excessive NET formation may be detrimental by inducing inflammation, tissue damage and mucociliary perturbation[29]. Higher levels of NET complexes have been reported in COPD frequent exacerbators[16]. Accordingly, we hypothesised that NETs may be an important driver of exacerbation pathogenesis in COPD. Our data from two different human study settings (experimentally-induced and naturally-occurring exacerbations) strongly support this assertion. We found that airway NETs measured by complementary immunoassays (quantification of DNA-elastase complexes and total/citrullinated nucleosomes) were induced by viral infection to a greater extent in COPD compared to healthy smoker or non-smoker control donors. Moreover, these markers correlated with virological (virus loads), inflammatory (cytokines, airway cells, mucin concentrations) and clinical (symptoms, acute lung function decline) measures of exacerbation severity. Our findings are consistent with a previous study of naturally occurring COPD exacerbations where immunodetection of NETs by confocal laser microscopy revealed increased rates in 28 participants with stable COPD (45% versus <5% in non-smokers) with further augmentation in 16 participants with exacerbated disease (over 90%)[17].

Using administration of a neutrophil elastase inhibitor in a preclinical model of virus-exacerbated disease we identified a functional role for NETs in driving immunopathology by showing that this intervention reduces inflammation, mucus secretion and AHR in mice. Systemic neutrophil elastase targeting therapies have been trialled in COPD and a range of other respiratory conditions including cystic fibrosis and non-CF bronchiectasis[30–32]. These approaches have failed to show efficacy upon clinical endpoints including symptoms and lung function which may be due to lack of target engagement in the lungs as no effect was observed on neutrophil counts or neutrophil elastase activity. These studies were conducted in the context of stable disease and the data from our study indicate that acutely targeting neutrophil elastase/NETs during the early phases of viral exacerbations may have more profound impacts. Alternative targeted approaches such as inhaled formulations of neutrophil elastase inhibitors are currently under evaluation (e.g., NCT04010799) and may also prove to be more effective. Other agents such as the DPP-1 inhibitor brensocatib have shown clearer attenuating effects upon sputum neutrophil activity with associated clinical efficacy in terms of exacerbation prevention in bronchiectasis[33] but these approaches have not yet been tested in COPD.

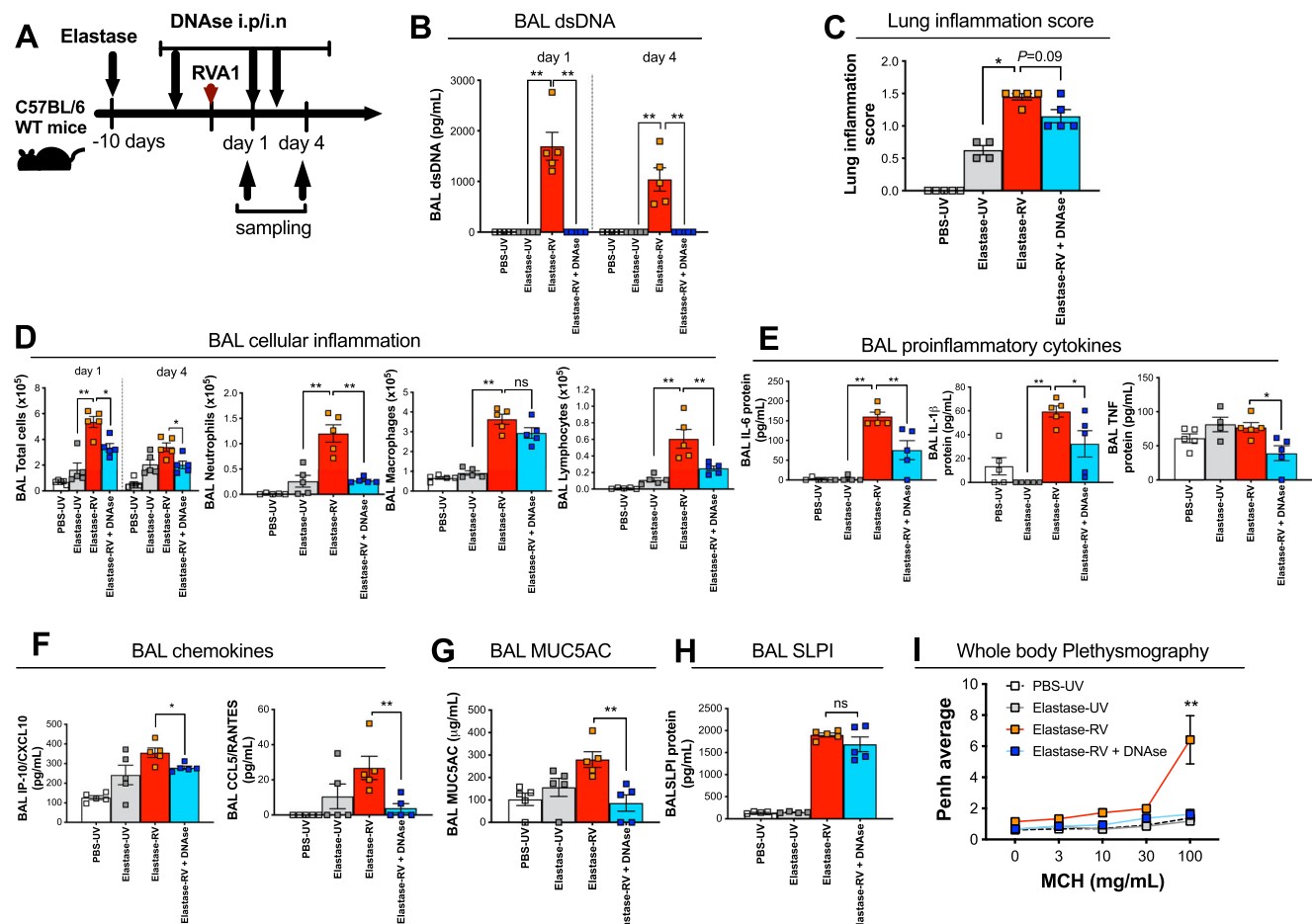

**Fig. 7 | DNAse treatment reduces immunopathology in a mouse model of virus-exacerbated COPD. A** Experimental schematic. Mice were treated by DNaseI (or vehicle control) by i.p. injection 4 h before inoculation and 1 and 2 days post-infection and by the intranasal route 8 h and 1 and 2 days post-infection (**B**) Bronchoalveolar lavage (BAL) concentrations of dsDNA was quantified by ELISA. **C** Histological lung inflammation scores (**D**) BAL total cells, neutrophils, macrophages and lymphocytes were enumerated by cytospin. BAL concentrations of (**E**) chemokines CXCL10/IP-10 and CCL5/RANTES and CCL2 (**F**) pro-inflammatory cytokines TNF, IL-1β and IL-6 and (**G**) mucin glycoprotein Muc5ac and (**H**) secretory leucocyte protease inhibitor (SLPI) were quantified by ELISA. **I** Airway hyperresponsiveness (enhanced pause [Penh]) to methacholine challenge was measured by whole body plethysmography. *n* = 4–5 mice/group representative of at least two independent experiments. Data are presented as mean values ± SEM. Data analysed by two-tailed Mann–Whitney U test. *P < 0.05, **P < 0.01,. Source data are provided as a Source Data file.

To dissect the mechanisms through which NETosis may drive immunopathology during COPD exacerbations, we further focussed on evaluation of dsDNA, a major NET constituent that is released into the airways in conjunction with NET formation. We have previously shown that dsDNA is a central driver of asthma exacerbations specifically through stimulation of type 2 inflammation[15]. We reasoned that dsDNA, which is widely recognised to act as a DAMP[34], would similarly induce inflammation and drive immunopathology in the context of COPD exacerbations. Accordingly, sputum dsDNA was induced by experimental RV infection to a greater extent in participants with COPD, peaking at the same timepoint as observed for NET formation (day 9 post-infection). This dynamic is consistent with our interpretation that NET formation leads to dsDNA release which, in turn, drives virological, immunological, and clinical exacerbation severity. Accordingly, we found that specific targeting of NETS in mice using a PAD inhibitor led to attenuation of dsDNA induction by RV, providing direct causal evidence that virus-induced airway host dsDNA is released from NETs during acute infection. DNAse administration in the preclinical model of virus-exacerbated COPD potently diminished inflammation, mucus secretion, and airways hyper-responsiveness. DNAse is well recognised to have mucolytic effects and has been shown to reduce exacerbations in cystic fibrosis[35] Conversely this

intervention led to increased exacerbations and accentuated FEV1 decline when evaluated in non-CF bronchiectasis[35]. One hypothesis for this difference relates to the fact that the underlying defect in CF is abnormally viscous/tenacious mucus which impairs ciliary clearance. In non-CF bronchiectasis (and also likely in COPD), the mucus is more normal and patients tend to have less difficulty clearing it unless they develop advanced disease that impairs cough efficacy. Therefore, it has been postulated that therapies aimed at reducing mucus viscosity and increasing mucus water content might not be useful in non-CF chronic respiratory conditions such as COPD[36]. Importantly, prior studies have examined daily maintenance administration of DNAse in patients with CF and non-CF bronchiectasis. Our data raises speculation that acute short term use of DNAse during virus-exacerbated disease may be beneficial in COPD and potentially other chronic lung conditions. Further interventional studies in humans will be required to further evaluate this. A recent meta-analysis of add-on mucolytic therapy in COPD exacerbations found increased rates of treatment success[37], although no studies to date have specifically evaluated DNAse in this context. We have previously shown that targeting early mucin production using an epidermal growth factor receptor antagonist reduces immunopathology in the same mouse model of exacerbated disease[20] and in the current study we have similarly shown

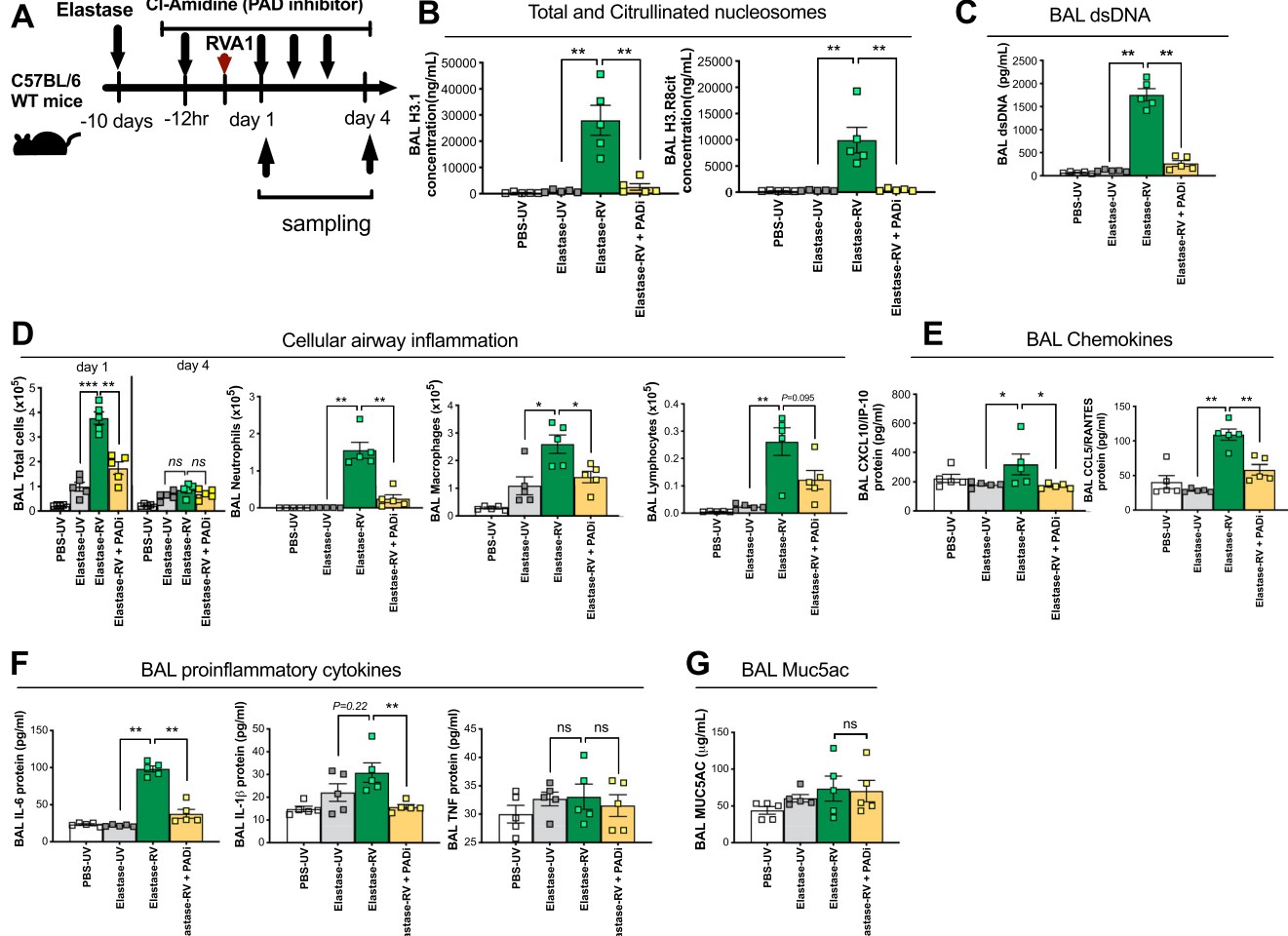

**Fig. 8 | Pharmacological inhibition of NETosis using a PAD inhibitor reduces dsDNA and immunopathology in a mouse model of virus-exacerbated COPD. A** Experimental schematic. Mice were treated with intranasal porcine pancreatic elastase or PBS control. 10 days later, mice received i.p. injection of a PAD inhibitor PADi; BB-CL-Amidine or vehicle control 12 h before RVA1 inoculation. **B** Bronchoalveolar lavage (BAL) concentrations of Total intact (H3.1) and citrullinated (H3.R8) nucleosomes were quantified by ELISA. **C** BAL concentrations of dsDNA quantified by ELISA. **D** BAL total cells, neutrophils, macrophages and lymphocytes. BAL concentrations of (**E**) chemokines CXCL10/IP-10 and CCL5/RANTES (**F**) pro-inflammatory cytokines IL-6, IL-1β,TNF and (**G**) Muc5ac were quantified by ELISA. $n = 5$ mice/group. Data expressed as mean ± SEM. Data analysed by two-tailed Mann–Whitney U test. *$P < 0.05$, **$P < 0.01$, ***$P < 0.001$. Source data are provided as a Source Data file.

that DNAse may also have beneficial effects during virus-induced exacerbations.

Both neutrophil elastase inhibition and DNAse administration led to similar attenuation of virus-induced inflammation. Our data from preclinical models would slightly favour the use of neutrophil elastase or NET-targeting therapies over DNAse administration, since the former led to beneficial modulation of SLPI, an antimicrobial peptide that is degraded by neutrophil elastase during COPD exacerbations to promote secondary bacterial infections[38] However, the direct implications of this effect requires formal testing using virus-bacterial co-infection mouse models and ultimately human intervention studies will be required for definitive evaluation. Several methodologies are available for quantification of NETs including immunolabelling, electron microscopy and flow cytometric approaches[29]. We applied flow cytometry to measure CXCR4 and Lamp1-expressing neutrophils within our mouse model but could not evaluate this within our human studies which were conducted on banked samples. In the current study, we therefore employed immunoassays measuring NET complexes (DNA–elastase) and citrullinated histones. These approaches measure surrogates of NET formation and it should be noted that, although broadly similar findings were observed in our controlled

human challenge study, within the real-world exacerbation study (where there is greater heterogeneity in disease severity, comorbidities, treatments and timing of sampling from exacerbation onset), sputum DNA-elastase concentrations fell from exacerbation onset to 2 weeks whilst sputum H3.R8Cit concentrations showed the opposite effect. Future studies should employ real time methods prospectively to further validate our fundings.

The pathogenic role of NETs during respiratory viral infections has garnered major interest recently in the context of COVID19 where high NET concentrations have been associated with the development of acute respiratory distress syndrome, tissue injury, and adverse outcomes[10,39], complications which occur more frequently in higher risk individuals. These findings mimic ours in the context of rhinovirus infection in COPD. We have also previously shown that NETs play a mechanistic role as a driver of post-COVID pulmonary sequelae[9]. In the current study, we did not evaluate longer term implications of rhinovirus infection beyond 42 days post-infection. It is feasible that higher induction of NETs may similarly lead to delayed symptomatic recovery. Studies with longer term follow-up will be required to further investigate this.

The elastase model of COPD, when exacerbated by RV infection, recapitulates several immunopathological features of human COPD

exacerbation including augmented airway inflammation, mucus hypersecretion, and lung function abnormalities. However, existing animal models including elastase or cigarette smoke exposure, are unable to fully reproduce the complex heterogenous disease features of human COPD, which occurs in genetically susceptible individuals following prolonged cigarette smoke exposure. Our study focussed primarily on rhinovirus infection as the most common viral trigger for COPD exacerbations. However, a range of other viruses including influenza and respiratory syncytial virus may also induce exacerbations in COPD[1]. Future studies are required to evaluate whether the mechanisms identified in the current study are also relevant to infections with other viruses.

In conclusion, our study uncovers a mechanistic role for neutrophils, NETs, and dsDNA in driving virus-induced exacerbations of COPD. Future development or repurposing of therapies that specifically target these pathways could represent an effective method of improving clinical outcomes.

## Methods

### Ethics statement
**Human experimental challenge study**. the study received ethical approval from St Mary's NHS Trust Research Ethics Committee (study number 07/H0712/138). Informed consent was obtained from all participants.

**Naturally occurring COPD exacerbation study**. the study received ethical approval from the East London Research Ethics Committee (study number 11/LO/0229). Informed consent was obtained from all participants.

**Animal experiments**. all animal experiments were performed under the authority of the UK Home Office outlined in the Animals (Scientific Procedures) Act 1986 after ethical review by Imperial College London Animal Welfare and Ethical Review Body (project licences 70/7234 and PP4051423).

### Human experimental challenge studies
Biobanked samples were analysed from a study of experimental rhinovirus challenge in persons with COPD (Global Initiative for Obstructive Lung Disease [GOLD] stage II), healthy smokers or healthy non-smokers, as previously reported[3]. Stable-state sputum samples were acquired 14 days prior to RV challenge, in people who had all had at least 6 weeks without any respiratory infection. Participants were inoculated intranasally with 10 TCID50 rhinovirus A16 at day 0. Patients were followed up sequentially with clinical monitoring and sample acquisition at day 3,5,9,12,15,21 and 42 post-infection. Symptom diary cards of upper and lower respiratory tract symptom, scores and peak expiratory flow (PEF) measurements were taken daily as previously described[3].

### Naturally occurring COPD exacerbation studies
Biobanked samples were analysed from a prior study examining the pathogenesis of naturally occurring viral exacerbations of COPD, conducted at St. Mary's Hospital London between June 2011 and December 2013, as described previously[23,40]. All patients had a confirmed clinical diagnosis of COPD with varying degrees of severity. All therapeutic interventions were permitted. Participants were clinically assessed at baseline (stable-state), where clinical assessment and collection of sputum samples were carried out. Upon development of respiratory tract symptoms, sputum samples were collected within 48 h of onset of symptoms (exacerbation onset). Sputum sample collection and clinical assessment were repeated at two weeks after exacerbation onset. Viral infection was confirmed by PCR detected at onset of infection.

### Mouse models
Female mice (age 6–8 weeks) on a C57BL/6 background were purchased from Charles River Laboratories UK. In all studies, mice were housed in individually ventilated cages under specific pathogen free conditions. Mice had access to food and water ad libitum with 12 h alternating light/dark cycles at temperature of 20–24 °C.

**Elastase-induced model of COPD**. For induction of a COPD-like phenotype, mice were treated intranasally under light isofluorane anaesthesia with 1.2 units of porcine pancreatic elastase (Merck, UK) as previously described[21] and left for ten days before treatment/infection as described below.

**In vivo treatment with neutrophil elastase inhibitor**. Neutrophil elastase inhibitor was administered as previously described[15]. Mice were injected intraperitoneally with 50 µg of neutrophil elastase inhibitor (GW311616A, AxonMedChem, Groningen, Netherlands) or vehicle control 12 h before inoculation with RV-A1 and at every 12hr interval until end-point analyses.

**DNase I treatment of mice**. DNase I was administered as previously described[15]. Mice were injected intraperitoneally with 1000 IU of DNase I from bovine pancreas (Sigma-Aldrich) in 200 µl of PBS 4 h prior to RV-A1 with further dose administration at 24 and 48 h after infection. Mice were concomitantly treated intranasally with 500 IU DNase I in 50 µl PBS at 8 h, day 1 and day 2 post-infection.

**In vivo treatment with BB-Cl-Amidine**. BB-Cl-Amidine, a modified version of the PAD-inhibitor Cl-Amidine (with 20 fold increased activity against PAD4, Cayman Chemicals) was administered as previously described[22]. Mice were injected intraperitoneally with 200 µg of BB-Cl-Amidine or vehicle control 12 h before inoculation with RV-A1 and at every 12 h interval until end-point analyses.

**Rhinovirus inoculation**. 10 days after elastase administration, mice were infected intranasally with 50 µl RV-A1 ($2.5 \times 10^6$ TCID$_{50}$) or UV-inactivated RV-A1 control.

**Whole-body plethysmography**. Airways hyper-responsiveness to nebulised methacholine challenge (measured as enhanced pause or PenH) was assessed using unrestrained whole-body plethysmography (Electromedsystems), as previously described[41].

### Cytospin
Mouse BAL cells were pelleted by centrifugation, treated with ammonium-chloride-potassium (ACK) buffer to lyse red blood cells and resuspended in RPMI medium with 10% FBS. Cells were then spun on to slides and stained with Quik-Diff (Reagena) for differential counts. Counts were performed blinded to experimental conditions.

### Mouse lung tissue preparation
The left lung lobe was removed, manually disaggregated with scissors, and incubated at 37 °C for 30 min in RPMI 1640 with 10% FBS (R10F) containing 0.15 mg/mL Liberase (Roche Diagnostics) and 25 µg/mL DNase (Roche Diagnostics) under gentle agitation. The lung tissue digest was filtered through a 70 µm sieve and centrifuged at $800 \times g$ for 5 min. Red blood cell lysis was performed by the addition of ACK buffer (0.15 M ammonium chloride, 1 M potassium hydrogen carbonate and 0.01 mM EDTA, pH 7.2) for 3 min at room temperature, before centrifugation at $800 \times g$ for 5 min. Supernatant was removed and the remaining cell pellet was resuspended in 2 mL R10F for counting.

For histology, lungs were perfused with PBS via the heart and inflated with 4% paraformaldehyde (PFA), then immersion fixed in 4%

PFA for 18 h. Samples were embedded in paraffin wax and 5-μm-thick histological sections were cut and stained with haematoxylin and eosin (H&E). The severity of inflammatory response observed in the stained lung sections was scored on a 0–3 scale, as previously reported[42].

## Flow cytometry

Lung single cell suspensions were stained with LIVE/DEAD Fixable Near IR Dead cell staining kit (Invitrogen) for 10 min at room temperature in the dark. Stained cells were washed in flow cytometry buffer (PBS + 0.1% sodium azide + 1% bovine serum albumin [BSA]) and centrifuged at 800 × $g$ for 5 min. Cells were then stained with either CD45-BV711, CD11c-APC (BD Pharmingen), CD11b-PerCP, SiglecF-BV421 (BD Pharmingen), Ly6G-FITC, Ly6C-Alexa Fluor 700, F4/80-PE, CD63 PE-Dazzle594, CD64-PE-Cy7, Fc Block (BD Bioscience), or CD45-Alexa Fluor 700, CD11b-PerCP, SiglecF-BV421 (BD Pharmingen), Ly6G-FITC, CXCR4-PE (BD Biosciences), CD47-PE-Dazzle594, CD49d-BV711 (BD Biosciences), Lamp-1-PE-Cy7, Fc Block (BD Bioscience) for 30 min at 4 °C in flow cytometry buffer and then fixed with 1% paraformaldehyde. Samples were acquired on a BD LSR Fortessa III cell analyser (BD Biosciences) and analysed using FlowJo version 10.9.0 (BD Biosciences). All antibodies were purchased from BioLegend unless otherwise stated. A representative gating strategy is shown in Supplementary Fig. 11.

## Measurement of total H3.1 and NET markers

Sputum concentrations of DNA-elastase complexes were quantified as previously described[16]. Anti-DNA primary antibody (HYB331-01, Abcam) was used to coat plates overnight, followed by wash steps and blocking with 1% BSA. A standard curve was generated through titration of healthy human blood-derived neutrophils stimulated with phorbol 12-myristate 13-acetate (PMA). Samples and standards were incubated followed by detection with sheep anti–neutrophil elastase–horseradish peroxidase (PA1-74133; Thermo Scientific) and developed with 3,3′5,5′-tetramethylbenzidine. Human sputum or mouse BAL concentrations of H3.1 and H3.R8Cit were quantified using the Volition Nu.Q H3.1 ELISA and Nu.Q H3.R8Cit ELISA assay kits, respectively, according to the manufacturer's instructions and as previously described[9,43].

## Measurement of dsDNA

dsDNA was quantified in the acellular fraction of sputum supernatants (human studies) or BAL fluid (mouse studies) using Quant-iT Pico-Green dsDNA reagent (Invitrogen, Carlsbad, CA), according to the manufacturer's instructions and as previously described[15].

## Measurement of cytokines and mucins

Cytokine protein concentrations in human sputum were quantified using the MesoScale Discovery (MSD) platform according to the manufacturer's instructions, as previously described[3]. Cytokine proteins in mouse BAL fluid or human sputum were quantified using commercial duoSet ELISA kits (R&D Systems). BAL lactate dehydrogenase (LDH) concentrations were quantified using the Pierce LDH cytotoxicity assay kit (Thermoscientific). MUC5AC protein concentrations in human sputum or mouse BAL were quantified using an in house assay, as previously described[20].

## RNA extraction, RT-PCR and qPCR

For mRNA expression analysis in mice, the right upper lobe was excised and total RNA was extracted using an RNeasy Mini kit (Qiagen, Valencia, CA). cDNA was subsequently synthesised (Omniscript RT Kit, QIAGEN). qPCR was conducted using specific primers and probes to quantify RV RNA copy numbers and normalised to 18 S, as previously described[44].

## Statistical analysis

Data from the human experimental challenge and naturally occurring viral exacerbation studies were analysed using the Wilcoxon matched-pairs, signed-rank test (paired data) or two-tailed Mann–Whitney $U$ test (unpaired data). Spearman's rank correlation coefficient was used for all correlation tests. All animal experiments were conducted using group sizes of 4–5 mice per group, and analysed using a two-tailed Mann–Whitney $U$ test. All statistical tests, correlations and analyses were performed using GraphPad Prism 10 (GraphPad Software). Differences were considered significant when the $P$ value was less than 0.05.

## Reporting summary

Further information on research design is available in the Nature Portfolio Reporting Summary linked to this article.

## Data availability

These data supporting the findings of the study are available in this article and its supplementary Information files, or from the corresponding author on request. Source data are provided with this paper.

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

## Acknowledgements

For the purpose of open access, the author has applied a Creative Commons Attribution (CC BY) licence to any Author Accepted Manuscript version arising. The authors acknowledge financial support from Imperial College London through an Imperial College Research Fellowship grant awarded to K.T.M. R.J.S. is a Wellcome Trust Senior Research Fellow in Basic Biomedical Sciences (209458/Z/17/Z). Parts of this work were also funded through a Rosetrees Trust/The Stoneygate Trust Project Grant (PGS21/10072). S.L.J. is a National Institute for Health Research (NIHR) Emeritus Senior Investigator and received support from the Asthma UK Clinical Chair (Grant CH11SJ), European Research Council Advanced Grants 233015 and 788575, Medical Research Council Centre Grant G1000758 and Asthma UK Centre Grant AUK-BC-2015-01. J.D.C. is supported by the Asthma and Lung UK Chair of Respiratory Research. A.S. is supported by an MRC Clinician Scientist Fellowship (MR/V000098/1).

## Author contributions

P.M., J.F., S.L.J., and A.S. conceived, designed, and analysed the human studies. O.K., M.T., and R.J.S., and A.S. conceived, designed and analysed the animal experiments. P.M., J.F., and T.K. performed sampling and experimental work related to human in vivo studies. O.K., M.T., M.M.J., K.T.M., G.F.M.M., and A.S. performed experimental work related to the animal studies. A.G., M.L., A.D.A., and J.D.C. coordinated and analysed NET assays. O.K., M.M.J., A.D.A., R.J.S., S.L.J., J.D.C., and A.S. contributed to writing and critical review of the manuscript.

## Competing interests

S.L.J. has personally received consultancy fees from AstraZeneca, Bioforce, Enanta and GlaxoSmithKline. S.L.J. is an inventor on patents on the use of inhaled interferons for treatment of exacerbations of airway diseases and on rhinovirus vaccines. S.L.J. is Director and shareholder of Virtus Respiratory Research Ltd. JDC has received research grants from AstraZeneca, Boehringer Ingelheim, GlaxoSmithKline, Gilead Sciences, Grifols, Novartis, Insmed and Trudell; and received consultancy or speaker fees from Antabio, AstraZeneca, Boehringer Ingelheim, Chiesi, GlaxoSmithKline, Insmed, Janssen, Novartis, Pfizer, Trudell and Zambo. A.S. has received honoraria for speaking from AstraZeneca. A.D.A. is Chief Medical Officer at Santersus AG. The remaining authors declare no competing interests.
