## [Peer Review File · Nature Communications]

Neutrophil extracellular traps promote immunopathogenesis of virus-induced COPD exacerbationsREVIEWER COMMENTS

Reviewer #1 (Remarks to the Author):

Reviewer's response

The manuscript written by Katsoulis et al titled "Neutrophil extracellular traps drive immunopathogenesis of virus-induced COPD exacerbations" studied the role of NETs in COPD pathology using both, human samples and mouse models. The work studies an important and understudied clinical problem, COPD pathogenesis, and proposes a significant role of NETs and RV infection in the process. Neutrophils and NETs have already been detected in the airways of COPD patients but mechanistic studies are missing that would address the role of neutrophils in COPD pathogenesis. Thus, the current study aims at filling in an important gap in our understanding of this topic. The human data are impressive enhancing the human clinical relevance of the work and the mouse experiments enabled a mechanistic approach. Overall, the data are interesting and support a role of neutrophils in COPD disease pathogenesis but the results are overinterpreted and do not fully prove the pathogenic role of NETs in the process, as the title says it. Additional experiments are needed to accurately prove the mechanistic role of NETs in COPD. Please find my major and minor comments below.

Major comments

- From the total histone 3 and citrullinated histone 3 results of Figure 1 the authors conclude that NET formation is upregulated during virus-induced COPD exacerbation. While the NET-marker, citH3 increased during the course of the viral infection, so did the level of total histone 3, as well. These data suggest that these increases are not necessarily specific to NETs but most likely to an enhanced neutrophil activation/recruitment. Could the authors gather more data to better dissect this question by measuring total NE, MPO in the sputum samples that are markers of neutrophil activation but are not NET-specific during the course of infection? Or any other measure to be assessed?
- please stop referring to total histone H3 as a NETosis marker when it is not. Histone release from neutrophils can occur by any cell death mechanism and it is not specific to NET extrusion (unlike citrullinated H3 or NE-DNA complex). Therefore, related conclusions made

from the total H3 data regarding correlation of NETs with inflammatory markers (Fig 2) should also be rephrased.

- line 158-167 (Fig 3 data). There is a discrepancy between the two NET markers studied during naturally occurring COPD exacerbations. DNA-NE complex peaks at the onset of exacerbation and returns to baseline 2 weeks after that, while citH3 is significantly highest in the last sample. Both markers detect NETs. How is it possible that when one NET marker (DNA-NE) peaks, the other one is non-significantly elevated and vice versa? Could additional NET markers be applied to clarify this question? The authors should elaborate on possible scenarios.

- GW311616A is an NE inhibitor, it is not a NET inhibitor as the authors state it - even if NE has been implicated in NET formation. The inhibitor could exert its effects on NE and consequently on lung disease, completely independent of NETs. In an attempt of the authors to connect NETs to lung pathology, inhibitors of PAD4, the enzyme mediating histone citrullination, provide the best, most NET-specific approach. Or PAD4-deficient mice could be tested in their NE+RV model to see whether lung pathology is decreased. Data in figure 5 prove that NE (and not NETs) drive lung disease in their model which is expected based on data and also the use of NE in the model. I would rephrase the title of figure 5 according to this and perform additional experiments related to PAD4 to conclude about the role of NETs.

- line 231: "likely released from NETs". While it is likely, the authors need to prove that dsDNA originates from NETosing neutrophils in their model. Please measure dsDNA levels when neutrophils are depleted in the mouse model or when NETs are not produced by inhibiting PAD4 (pharmacologically or genetically, as detailed above).

- Figure 7 human data. The fact that dsDNA correlates with histones does not prove that dsDNA is derived from NETs, as dsDNA levels correlated with everything else the authors measured including cytokines and mucins, as well.

Minor comments

- line 85-86. Myeloperoxidase is not a neutrophil protease. Please rephrase it.

- line 93. Please clarify the meaning of this part of the sentence: "Although normally sequestered from potential signal receptors...". How is dsDNA sequestered by signal receptors?

- DNase treatment has been prescribed to cystic fibrosis patients for decades and it somewhat eases their lung disease that is characterized by chronic neutrophilic inflammation including NETs. Could the authors discuss why DNase therapy has not been studied or approved in COPD, a disease representing the third leading cause of death worldwide affecting a much larger human population than CF?

Reviewer #2 (Remarks to the Author):

This manuscript by Katsoulis et al. investigated the roles of neutrophil extracellular traps (NETs) and NET-released double-stranded DNA (dsDNA) in rhinovirus infection induced exacerbation of chronic obstructive pulmonary disease (COPD). The authors used human experimental infection, naturally-occurred virus-associated exacerbation, and also mouse in vivo experimental model of COPD and demonstrated that rhinovirus infection induces airway NET formation in these COPD models, and the markers of NETs formation including total intact (H3.1) and citrullinated (H3R8cit) nucleosomes were all highly increased by rhinovirus infection in COPD and correlated with immunological and clinical exacerbation measurements including sputum inflammatory cytokines/chemokines and lower respiratory tract symptom scores. The authors then showed that treatment of COPD mouse model with a specific NETosis inhibitor and a DNA-hydrolysing enzyme DNase I reduced immunopathology and disease severity in rhinovirus-induced exacerbation, indicating that NETs are important drivers of exacerbation and as potential therapeutics in COPD.

This study provides novel insight in the mechanisms of rhinovirus-induced exacerbation of COPD and assesses potential therapeutics. While the study is mostly methodologically sound, there are important experimental and clinical questions that need to be addressed.

Specific comments

This study only recruited those with COPD stage II, one would assume that NETosis would be more highly induced in stage III and IV but have the authored assessed the NET parameters in these groups?

The number of COPD individuals monitored in the naturally occurring exacerbation should also be stated in the main text.

Fig 3A, what does the big “I” between stable state and exacerbation presentation mean in the experiment schematic fig?

Page 5, “Sputum H3.1 showed trends towards increases at exacerbation onset and 2 weeks that failed to reach statistical significance (Fig 3B).” This should be Fig 3C, and please insert Fig 3B in the sentence prior to this.

Page 5, “Having observed that NETs are increased during virus-induced COPD exacerbations in man,”, please consider using “in the human experimental infection experiment or other similarly worded phrase” to be more inclusive and precise, instead of “in man”

The authors measured total neutrophils, CD63+ and CD64+, and CXCR4 and LAMP1 neutrophils in the COPD mouse model of infection, were these neutrophils also assessed in the human COPD sputum via IHC and compared with the non-COPD and smokers group?

On page 6, “RVs are RNA viruses with no dsDNA intermediates”, please provide a reference.

On page 6, spelling - “GW311616A, a neutrophil elastase inhibitor”

In all the experiments performed (both human and mouse in vivo), it would be beneficial to also assess the productions of important antiviral cytokines including type I – III interferons, and the effects of both NETosis inhibitor and DNase have on these cytokines. Also how would these therapeutics impact the disease severity on secondary viral or bacterial infections that are commonly observed in the discussion.

Given the both NETosis inhibitor and DNase are claimed and assessed as potential therapeutics, one would be tempted to ask which one is more effective in reducing disease severity and would inhibition of NETosis lead to longer term persistence (or accumulation) of neutrophils and NETs in the airways ?

Are the mouse model of COPD also steroid-resistant and have the authors assessed if

NETosis inhibitor and DNase restore sensitivity and if they could be used as a combined therapy with steroids ?

Have the authors also assessed these inhibitors against other common exacerbation-inducing viruses such as influenza viruses in mouse models? Mouse lung histopathology via HE and PAS staining would also provide additional data to show if the therapeutics reduce or prevent extensive lung damage caused by the virus.

The authors stated that dsDNAs detected in the airways are solely-host derived, please also discuss if the dsDNAs are only released by the neutrophils or are they also likely to be released from damaged epithelial cells undergoing necrosis due to virus infection ? mouse lung immunohistopathological staining/scores would somewhat provide clues to this, and the increased dsDNA observed might be due to both NETosis and damaged epithelial cells. However would the DNase treatment render the host unable to respond appropriately to secondary bacterial or viral infection?

Reviewer #3 (Remarks to the Author):

The authors examined the role of NETs in pulmonary inflammation using a mouse elastase model and in human samples from COPD patients.

Comments:

I think the manuscript is more suitable for a specialized lung journal. As the authors rightly note, the role of NETs in pulmonary inflammation has already been studied in mouse models, also with rhinovirus (e.g. PMID: 28459437, 36495695). Human samples from COPD patients have also been extensively analyzed for NETs (e.g. PMID: 28506850, 25994149).

The elastase model carried out in this study (which is actually more of a lung regeneration model) and the relatively small number of patient samples analyzed therefore only represents an extension of previous studies.

Fig.1: Is it really surprising that COPD patients have elevated "markers" for NETs on one (day 9) of 42 days, or does this just reflect an increased number of neutrophils in the sputum of COPD patients (as shown in Fig. 6)? Why are the results actually torn apart?

Fig.3.: Don't elevated markers for NETs simply reflect an increased number of leukocytes in

the sputum? Aren't numerous inflammatory factors increased?

The number of mice with n=5 is rather small. Have the experiments been repeated? The Elastase model is actually not a true COPD model. The fact that viruses lead to more markers for NETs in the model is not surprising given the massive neutrophilic inflammation in virus/elastase experiments. It is not clear to me how a role of NETs in pulmonary inflammation can be concluded in the model using an inhibitor of neutrophilic elastase given the massive influx of damaging elastase expressing neutrophils. In addition, the lungs were not examined for structural changes and pulmonary function, as is usually the case in such models.

Response to reviewers

Reviewer #1 (Remarks to the Author):

Reviewer's response

C1: The manuscript written by Katsoulis et al titled “Neutrophil extracellular traps drive immunopathogenesis of virus-induced COPD exacerbations” studied the role of NETs in COPD pathology using both, human samples and mouse models. The work studies an important and understudied clinical problem, COPD pathogenesis, and proposes a significant role of NETs and RV infection in the process. Neutrophils and NETs have already been detected in the airways of COPD patients but mechanistic studies are missing that would address the role of neutrophils in COPD pathogenesis. Thus, the current study aims at filling in an important gap in our understanding of this topic. The human data are impressive enhancing the human clinical relevance of the work and the mouse experiments enabled a mechanistic approach

R1: We thank the reviewer for the time spent reviewing the manuscript and agree with their assessment that the study fills an important gap and enhances human clinical relevance with use of mouse experiments to enable a mechanistic approach.

C2: Overall, the data are interesting and support a role of neutrophils in COPD disease pathogenesis but the results are overinterpreted and do not fully prove the pathogenic role of NETs in the process, as the title says it. Additional experiments are needed to accurately prove the mechanistic role of NETs in COPD. Please find my major and minor comments below.

R2: We thank the reviewer for these comments and agree regarding the need for additional experiments to accurately prove the mechanistic role of NETs in COPD. We have carried these out, as detailed in our responses to specific comments below.

C3: From the total histone 3 and citrullinated histone 3 results of Figure 1 the authors conclude that NET formation is upregulated during virus-induced COPD exacerbation. While the NET-marker, citH3 increased during the course of the viral infection, so did the level of total histone 3, as well. These data suggest that these increases are not necessarily specific to NETs but most likely to an enhanced neutrophil activation/recruitment. Could the authors gather more data to better dissect this question by measuring total NE, MPO in the sputum samples that are markers of neutrophil activation but are not NET-specific during the course of infection? Or any other measure to be assessed?

R3: We thank the reviewer for these important comments. Total NE has previously been measured in the human experimental challenge study with increases coinciding with similar dynamics to NET expression (day 9, reported in PMID 25790167, Figure 1c). We have therefore not repeated these measurements. However, we have now measured sputum MPO in the experimental challenge study and both total NE and MPO in the naturally occurring exacerbation study and these data are included in new supplementary Figures 2 and 5 (pasted below). We did not observe clear virus induction of MPO in sputum in either study (post human RV challenge or stable to exacerbation state in the naturally occurring study). By contrast total NE is clearly induced following experimental RV challenge (PMID 25790167) and is also induced from stable-state to exacerbation onset in the naturally occurring COPD exacerbation study (new supplementary Figure 5B (pasted below).

Supplementary Figure 2: Sputum myeloperoxidase concentrations during human RV infection in COPD. (A) Experiment schematic. 9 subjects with COPD, 10 healthy smokers and 11 healthy non-smokers underwent sampling at baseline and the indicated time points after experimental RV-A16 challenge. (B) Sputum myeloperoxidase (MPO) was quantified by ELISA. Data analysed by Wilcoxon matched pairs, signed-rank test and two-tailed Mann-Whitney test.

Supplementary Figure 5: Sputum neutrophil elastase and myeloperoxidase during naturally occurring virus associated exacerbations. (A) Experiment schematic. 18 individuals with COPD were monitored prospectively. Sputum samples were taken during stable state (baseline), at presentation with an exacerbation associated with positive virus detection, and 2 weeks after exacerbation presentation. Sputum concentrations of (B) Neutrophil elastase (NE) and (C) myeloperoxidase (MPO) were measured at stable-state and following virus-induced exacerbation. ** $P < 0.01$. Data analysed by Wilcoxon matched pairs, signed-rank test

C4:- please stop referring to total histone H3 as a NETosis marker when it is not. Histone release from neutrophils can occur by any cell death mechanism and it is not specific to NET extrusion (unlike citrullinated H3 or NE-DNA complex). Therefore, related conclusions made from the total H3 data regarding correlation of NETs with inflammatory markers (Fig 2) should also be rephrased.

R4: Thank you for this comment, which is well taken. We have rephrased accordingly.

C5 line 158-167 (Fig 3 data). There is a discrepancy between the two NET markers studied during naturally occurring COPD exacerbations. DNA-NE complex peaks at the onset of exacerbation and returns to baseline 2 weeks after that, while citH3 is significantly highest in the last sample. Both markers detect NETs. How is it possible that when one NET marker (DNA-NE) peaks, the other one is non-significantly elevated

and vice versa? Could additional NET markers be applied to clarify this question? The authors should elaborate on possible scenarios.

R5: We thank the reviewer for the comment and we also noted the discrepant results. Both the H3-cit and DNA-elastase complex markers are increased from stable-state to exacerbation but for H3-cit levels appear to increase further at 2 weeks with the opposite effect observed for DNA-elastase complexes. Although both markers are surrogates of NETosis, the assays are quantifying different components. Our naturally-occurring study is a heterogeneous 'real world' study where exacerbation trigger may occur through viral infection, bacterial infection or a combination (it is impossible to definitively ascertain in this study type). Subjects in this study also had a range comorbidities and medication histories (including individuals on and off inhaled corticosteroids). Any or all of these confounders could affect the expression of the various NET surrogate markers we tested and contribute to the discrepant findings observed in the naturally occurring exacerbation setting. In our more controlled experimental model where subjects are better matched for severity/treatments, timing of sampling relative to exacerbation onset and do not have major comorbidities, the data observed for these two NET markers was more consistent. We have added data on sputum total NE which follows a similar pattern to DNA-elastase complexes and sputum MPO which shows trends toward increases at exacerbation with a fall at 2 weeks (again mimicking DNA-elastase complexes). We have added a more detailed explanation of the above to the revised discussion section.

'These approaches measure surrogates of NET formation and it should be noted that, although broadly similar findings were observed in our carefully controlled human challenge study, within the real-world exacerbation study (where there is greater heterogeneity in disease severity, comorbidities, treatments and timing of sampling from exacerbation onset), sputum DNA-elastase concentrations fell from exacerbation onset to 2 weeks whilst sputum H3.R8Cit concentrations showed the opposite effect.'

C6: GW311616A is an NE inhibitor, it is not a NET inhibitor as the authors state it - even if NE has been implicated in NET formation. The inhibitor could exert its effects on NE and consequently on lung disease, completely independent of NETs. In an attempt of the authors to connect NETs to lung pathology, inhibitors of PAD4, the enzyme mediating histone citrullination, provide the best, most NET-specific approach. Or PAD4-deficient mice could be tested in their NE+RV model to see whether lung pathology is decreased. Data in figure 5 prove that NE (and not NETs) drive lung disease in their model which is expected based on data and also the use of NE in the model. I would rephrase the title of figure 5 according to this and perform additional experiments related to PAD4 to conclude about the role of NETs.

R6: Thank you. We agree and have rephrased to 'NE inhibitor' throughout. As recommended, we have conducted additional experiments using a PAD4 inhibitor (BB CI-amidine). These are presented in a new Figure 8 within the revised manuscript (pasted below) and indicate that more targeted inhibition of NETs through PAD4 inhibition has similar effects to NE inhibition with reduced virus-induced cellular inflammation and pro-inflammatory chemokine/cytokine secretion. This further strengthens our conclusion that NETs are drivers of immunopathology during virus-exacerbated COPD.

Figure 8: Pharmacological inhibition of NETosis using a PAD4 inhibitor reduces extracellular and immunopathology in a mouse model of virus-exacerbated COPD. (A) Experimental schematic. Mice were treated with intranasal porcine pancreatic elastase or PBS control. 10 days later, mice received i.p. injection of a PAD-4 inhibitor (PAD4i; CL-Amidine or vehicle control) 12 hours before RVA1 inoculation. (B) Bronchoalveolar lavage (BAL) concentrations of Total intact (H3.1) and citrullinated (H3.R8) nucleosomes were quantified by ELISA. (C) BAL concentrations of dsDNA quantified by ELISA. (D) BAL total cells, neutrophils, macrophages and lymphocytes were enumerated by cytopsin. BAL concentrations of (E) chemokines CXCL10/IP-10 and CCL5/RANTES (F) pro-inflammatory cytokines IL-6, IL-1 β , TNF and (G) Muc5ac were quantified by ELISA. n=5 mice/group. Data expressed as mean \pm -SEM. Data analysed by two-tailed Mann Whitney U test. * P <0.05, ** P <0.01, *** P <0.001

C7: line 231: “likely released from NETs”. While it is likely, the authors need to prove that dsDNA originates from NETosing neutrophils in their model. Please measure dsDNA levels when neutrophils are depleted in the mouse model or when NETs are not produced by inhibiting PAD4 (pharmacologically or genetically, as detailed above).

R7: Thank you for this comment. We have now measured dsDNA in the PAD4 inhibitor experiment which shows that NET inhibition leads to reduced dsDNA concentrations and thus provides further support for the statement that dsDNA originates from NETosing neutrophils in our model. See Figure 8c (pasted below).

C8:- Figure 7 human data. The fact that dsDNA correlates with histones does not prove that dsDNA is derived from NETs, as dsDNA levels correlated with everything else the authors measured including cytokines and mucins, as well.

R8: Thank you. We agree and have reworded this to temper the statement and also added the data measuring dsDNA in the PAD4 inhibitor studies (as stated above), which more clearly supports this.

*'These data **suggested** that virus-induced airway host dsDNA **may be** released from NETs and accentuated in COPD subjects during acute infection'*

Minor comments

C9:- line 85-86. Myeloperoxidase is not a neutrophil protease. Please rephrase it.

R9: Thank you, we have rephrased.

C10:- line 93. Please clarify the meaning of this part of the sentence: "Although normally sequestered from potential signal receptors...". How is dsDNA sequestered by signal receptors?

R10: Thank you, we have removed this.

C11: DNase treatment has been prescribed to cystic fibrosis patients for decades and it somewhat eases their lung disease that is characterized by chronic neutrophilic inflammation including NETs. Could the authors discuss why DNase therapy has not been studied or approved in COPD, a disease representing the third leading cause of death worldwide affecting a much larger human population than CF?

R11: Thank you. Although DNase has shown efficacy in cystic fibrosis, this benefit has not extended to other conditions such as non-CF bronchiectasis where its use led to adverse effects including increased exacerbations (PMID: 9596315). One hypothesis for this difference relates to the fact that the underlying defect in CF is abnormally viscous/tenacious mucus which impairs ciliary clearance. In non-CF bronchiectasis (and also likely in COPD), the mucus is more normal and patients have less difficulty clearing it unless they develop advanced disease that impairs cough efficacy. Therefore, it has been postulated that therapies aimed at reducing mucus viscosity and increasing mucus water content might not be useful in non-CF

chronic respiratory conditions such as COPD (PMID: 30906533). Importantly, prior studies have examined daily maintenance administration of DNase in patients with CF and non-CF bronchiectasis and our data raises speculation that acute short-term use of DNase during virus-exacerbated disease may be beneficial in COPD and potentially other chronic lung conditions. We have added detail about these points to the revised discussion.

'These data were further supported by our finding that DNase administration in the preclinical model of virus-exacerbated COPD potentially diminished inflammation, mucus secretion and airways hyper-responsiveness. DNase is well recognised to have mucolytic effects and has been shown to reduce exacerbations in cystic fibrosis³⁵ Conversely this intervention led to increased exacerbations and accentuated FEV₁ decline when evaluated in non-CF bronchiectasis.³⁵ One hypothesis for this difference relates to the fact that the underlying defect in CF is abnormally viscous/tenacious mucus which impairs ciliary clearance. In non-CF bronchiectasis (and also likely in COPD), the mucus is more normal and patients have less difficulty clearing it unless they develop advanced disease that impairs cough efficacy. Therefore, it has been postulated that therapies aimed at reducing mucus viscosity and increasing mucus water content might not be useful in non-CF chronic respiratory conditions such as COPD³⁶. Importantly, prior studies have examined daily maintenance administration of DNase in patients with CF and non-CF bronchiectasis. Our data raises speculation that acute short-term use of DNase during virus-exacerbated disease may be beneficial in COPD and potentially other chronic lung conditions.'

Reviewer #2 (Remarks to the Author):

C12: This manuscript by Katsoulis et al. investigated the roles of neutrophil extracellular traps (NETs) and NET-released double-stranded DNA (dsDNA) in rhinovirus infection induced exacerbation of chronic obstructive pulmonary disease (COPD). The authors used human experimental infection, naturally-occurred virus-associated exacerbation, and also mouse in vivo experimental model of COPD and demonstrated that rhinovirus infection induces airway NET formation in these COPD models, and the markers of NETs formation including total intact (H3.1) and citrullinated (H3R8cit) nucleosomes were all highly increased by rhinovirus infection in COPD and correlated with immunological and clinical exacerbation measurements including sputum inflammatory cytokines/chemokines and lower respiratory tract symptom scores. The authors then showed that treatment of COPD mouse model with a specific NETosis inhibitor and a DNA-hydrolysing enzyme DNase I reduced immunopathology and disease severity in rhinovirus-induced exacerbation, indicating that NETs are important drivers of exacerbation and as potential therapeutics in COPD. This study provides novel insight in the mechanisms of rhinovirus-induced exacerbation of COPD and assesses potential therapeutics. While the study is mostly methodologically sound, there are important experimental and clinical questions that need to be addressed.

R12: We thank the reviewer for the time spent assessing our manuscript and agree regarding the important experimental and clinical questions, which we have addressed in our responses to the specific comments below.

Specific comments

C13: This study only recruited those with COPD stage II, one would assume that NETosis would be more highly induced in stage III and IV but have the authored assessed the NET parameters in these groups?

R13: Thank you for this comment. Although our human experimental challenge study did not recruit subjects more severe than stage II, the naturally occurring exacerbation study did have

some subjects in the higher severity category. Due to numbers included in this study, subdivision leads to small group sizes but we have plotted these stratified data and presented this in new Supplementary Figure 4 (pasted below). For sputum DNA elastase complexes, the data shows that induction occurs most clearly in the higher disease severity group but this is not observed for the histone markers. Due to small numbers, we have been careful to not over-state the significance of these data but have merely presented them in the manuscript as a supplementary figure.

Supplementary Figure 4: Airway NET expression during naturally occurring virus associated exacerbations in COPD subjects stratified according to GOLD severity class.

(A) Experiment schematic. 18 individuals with COPD were monitored prospectively. Subjects were stratified into those with GOLD stage III-IV disease (n=5) and those with GOLD stage I-II disease (n=13). Sputum samples were taken during stable state (baseline), at presentation with an exacerbation associated with positive virus detection, and 2 weeks after exacerbation presentation. Sputum concentrations of (B) DNA-elastase complexes, (C) total intact (H3.1) and (D) citrullinated (H3.R8) nucleosomes were measured at stable-state and following virus-induced exacerbation. Data analysed by Wilcoxon matched pairs, signed-rank test.

C14: The number of COPD individuals monitored in the naturally occurring exacerbation should also be stated in the main text.

R14: Thank you, we have added this.

C15: Fig 3A, what does the big "I" between stable state and exacerbation presentation mean in the experiment schematic fig?

R15: Thank you, this was just to denote that there would be a variable amount time elapsed between stable state visit and exacerbation visit but we have removed this to aid clarity.

C16: Page 5, “Sputum H3.1 showed trends towards increases at exacerbation onset and 2 weeks that failed to reach statistical significance (Fig 3B).” This should be Fig 3C, and please insert Fig 3B in the sentence prior to this.

R16: Thank you for pointing this out, we have corrected.

C17: Page 5, “Having observed that NETs are increased during virus-induced COPD exacerbations in man,”, please consider using “in the human experimental infection experiment or other similarly worded phrase” to be more inclusive and precise, instead of “in man”

R17: Thank you, we have reworded this.

C18: The authors measured total neutrophils, CD63+ and CD64+, and CXCR4 and LAMP1 neutrophils in the COPD mouse model of infection, were these neutrophils also assessed in the human COPD sputum via IHC and compared with the non-COPD and smokers group?

R18: Thank you for this comment. Some of these activation markers, including CD63, have been assessed in the human model within previously published studies from our group employing flow cytometry (PMID 23834268) and IHC (PMID: 32283204). Unfortunately, we do not have sufficient sample from the human studies analysed in the current manuscript to perform further evaluation of other markers of neutrophil heterogeneity.

C19: On page 6, “RVs are RNA viruses with no dsDNA intermediates”, please provide a reference.

R19: Thank you, we have added a reference.

C20: On page 6, spelling - “GW311616A, a neutrophil elastase inhibitor”

R20: Thank you, have corrected.

C21: In all the experiments performed (both human and mouse in vivo), it would be beneficial to also assess the productions of important antiviral cytokines including type I – III interferons, and the effects of both NETosis inhibitor and DNase have on these cytokines. Also how would these therapeutics impact the disease severity on secondary viral or bacterial infections that are commonly observed in the discussion.

R21: Thank you for this comment. We have previously attempted to measure type I and III interferons (IFNs) in sputum from the human studies and these proteins were undetectable. In the animal studies, we have added data on antiviral response and virus load in each of the models. We observe no effect of neutrophil elastase inhibitor (NEi) or DNase on type I IFN-alpha responses to RV (see data pasted below, included in revised manuscript). IFN-beta and lambda proteins were not detectable at the timepoints assessed in these experiments.

Supplementary Figure 7: Pharmacological neutrophil elastase inhibition has no impact upon antiviral immunity. Mice were treated with elastase 10 days prior to treatment with Neutrophil elastase inhibitor or vehicle control and infection with RV-A1. (A) Bronchoalveolar lavage (BAL) concentrations of IFN- α were quantified by ELISA. (B) Lung tissue rhinovirus RNA copies were quantified by qPCR. n=5 mice/group, one experiment representative of two independent experiments is shown with data expressed as mean \pm -SEM. *P<0.05. ns=non-significant

Supplementary Figure 9: DNase treatment has no impact upon antiviral immunity. Mice were treated with elastase 10 days prior to treatment with DNase or vehicle control and infection with RV-A1. (A) Bronchoalveolar lavage (BAL) concentrations of IFN- α were quantified by ELISA. (B) Lung tissue rhinovirus RNA copies were quantified by qPCR. n=5 mice/group, one experiment representative of two independent experiments is shown with data expressed as mean \pm -SEM. *P<0.05, ns=non-significant.

Our prior data in COPD would suggest that neutrophil elastase inhibition would protect against secondary bacterial infection as we have previously shown that neutrophil elastase-mediated cleavage of SLPI promotes secondary bacterial infection in COPD (PMID: 23024024). Therefore targeting neutrophil elastase would be expected to prevent SLPI degradation and boost anti-bacterial immunity after viral infection. We have carried out further analyses in the mouse models to directly assess this and, accordingly, found that Neutrophil elastase inhibition in exacerbated mice does indeed boost RV-induction of SLPI whilst DNase inhibition has no discernible effect. These data are pasted below (Figure 5H and 7H) and included in the revised manuscript.

Figure 5: Pharmacological neutrophil elastase inhibition reduces immunopathology in a mouse model of virus-exacerbated COPD. (A) Experimental schematic. Mice were treated with intranasal porcine pancreatic elastase or PBS control. 10 days later, mice received i.p. injection of the Neutrophil elastase inhibitor (NEI; GW311616A) or vehicle control 12 hours before RVA1 inoculation. (B) Bronchoalveolar lavage (BAL) concentrations of Total intact (H3.1) and citrullinated (H3.R8) nucleosomes were quantified by ELISA. (C) Histological lung inflammation scores. (D) BAL total cells, neutrophils, macrophages and lymphocytes were enumerated by cytospin. (BAL concentrations of (E) chemokines CXCL10/IP-10 and CCL5/RANTES (F) pro-inflammatory cytokines TNF, IL-1 β and IL-6, (G) Muc5ac and (H) secretory leucocyte protease inhibitor (SLPI) were quantified by ELISA. (I) Airway hyperresponsiveness (enhanced pause [Penh]) to methacholine challenge was measured by whole body plethysmography. $n=4-5$ mice/group. Data expressed as mean \pm -SEM. Data analysed by two-tailed Mann Whitney U test. * $P<0.05$, ** $P<0.01$, *** $P<0.001$

Figure 7 DNase treatment reduces immunopathology in a mouse model of virus-exacerbated COPD. (A) Experimental schematic. Mice were treated by DNaseI (or vehicle control) by i.p. injection 4 hours before inoculation and 1 and 2 days post-infection and by the intranasal route 8 hours and 1 and 2 days post-infection (B) Bronchoalveolar lavage (BAL) concentrations of dsDNA was quantified by ELISA. (C) Histological lung inflammation scores (D) BAL total cells, neutrophils, macrophages and lymphocytes were enumerated by cytopsin. BAL concentrations of (E) chemokines CXCL10/IP-10 and CCL5/RANTES and CCL2 (F) pro-inflammatory cytokines TNF, IL-1 β and IL-6 and (G) mucin glycoprotein Muc5ac and (H) secretory leucocyte protease inhibitor (SLPI) were quantified by ELISA. (I) Airway hyperresponsiveness (enhanced pause [Penh]) to methacholine challenge was measured by whole body plethysmography. n=4-5 mice/group. Data analysed by two-tailed Mann Whitney U test. * $P < 0.05$, ** $P < 0.01$, *** $P < 0.001$.

C22: Given the both NETosis inhibitor and DNase are claimed and assessed as potential therapeutics, one would be tempted to ask which one is more effective in reducing disease severity and would inhibition of NETosis lead to longer term persistence (or accumulation) of neutrophils and NETs in the airways ?

R22: Thank you for this comment. Based on the above data, we would suggest that Neutrophil elastase inhibition may be more effective, given its beneficial effect upon antibacterial immunity (SLPI). We have added this detail to the discussion but ultimately clinical trials of these agents in human exacerbations will be required (see pasted text below).

‘Both neutrophil elastase inhibition and DNase administration led to similar attenuation of virus-induced inflammation. Our data from preclinical models would slightly favour the use of neutrophil elastase-targeting therapies over DNase administration, since the former led to beneficial modulation of SLPI, an antimicrobial peptide that is degraded by neutrophil elastase during COPD exacerbations to promote secondary bacterial infections³⁷ However, the direct implications of this effect requires formal testing using virus-bacterial co-infection mouse models and ultimately human intervention studies will be required for definitive evaluation.’

With regards to longer term persistence or accumulation of neutrophils in the airways after NETosis inhibition, we have added additional data assessing BAL neutrophil counts at a later timepoint (day 4 post-infection) and see no evidence of either neutrophil elastase inhibition or PAD4 inhibition leading to increased neutrophils. These data are included in new supplementary figures 6 and 10 (pasted below).

Supplementary Figure 6: Effect of pharmacological neutrophil elastase inhibition upon neutrophil recruitment at day 4 post RV infection. BAL neutrophils were enumerated by cytospin in mice treated with elastase 10 days prior to infection with RV-A1 treated with Neutrophil elastase inhibitor or vehicle control. n=5 mice/group, one experiment representative of two independent experiments is shown with data expressed as mean+/-SEM.

Supplementary Figure 10: Effect of pharmacological NET inhibition upon neutrophil recruitment at day 4 post RV infection. BAL neutrophils were enumerated by cytospin in mice treated with elastase 10 days prior to infection with RV-A1 treated with PAD4 inhibitor (BB-CL-amidine) or vehicle control. n=5 mice/group, one experiment representative of two independent experiments is shown with data expressed as mean+/-SEM.

C23: Are the mouse model of COPD also steroid-resistant and have the authors assessed if NETosis inhibitor and DNase restore sensitivity and if they could be used as a combined therapy with steroids ?

R23: Thank you, the inflammation in the mouse model of RV exacerbated COPD is not steroid resistant, as shown in our previous work where pulmonary administration of fluticasone potentially inhibited neutrophilic inflammation and pro-inflammatory cytokine responses (PMID: 29884817). Other models that better recapitulate steroid insensitivity would be required to formally test this.

C24: Have the authors also assessed these inhibitors against other common exacerbation-inducing viruses such as influenza viruses in mouse models?

R24: Thank you. We have not tested these inhibitors against other viruses as the primary focus in our animal models was rhinovirus infection, as the commonest cause of COPD exacerbations and to provide direct comparison to our human experimental challenge model which utilises rhinovirus and not other viruses. We have added detail to the discussion that future studies may wish to confirm similar effects with other viral triggers.

'Our study focussed primarily on rhinovirus infection as the most common viral trigger for COPD exacerbations. However, a range of other viruses including influenza and respiratory syncytial virus (RSV) may also induce exacerbations in COPD¹. Future studies are required to evaluate whether the mechanisms identified in the current study are also relevant to infections with other viruses.'

C25: Mouse lung histopathology via HE and PAS staining would also provide additional data to show if the therapeutics reduce or prevent extensive lung damage caused by the virus.

R25: Thank you. We have added data on histological inflammation scores via H&E staining in the DNase and neutrophil elastase inhibition models and show that both these treatments reduce total inflammation scores, in keeping with our cellular data (see pasted figures - 5C and 7C below). Unfortunately we did not conduct PAS staining in these experiments although we do show significant effects of DNase in reducing BAL MUC5AC concentrations in Figure 7G, with a similar trend for neutrophil elastase inhibition in 5G.

Figure 5: Pharmacological neutrophil elastase inhibition reduces immunopathology in a mouse model of virus-exacerbated COPD. (A) Experimental schematic. Mice were treated with intranasal porcine pancreatic elastase or PBS control. 10 days later, mice received i.p. injection of the Neutrophil elastase inhibitor (NEI; GW311616A) or vehicle control 12 hours before RVA1 inoculation. (B) Bronchoalveolar lavage (BAL) concentrations of Total intact (H3.1) and citrullinated (H3.R8) nucleosomes were quantified by ELISA. (C) Histological lung inflammation scores. (D) BAL total cells, neutrophils, macrophages and lymphocytes were enumerated by cytospin. (BAL concentrations of (E) chemokines CXCL10/IP-10 and CCL5/RANTES (F) pro-inflammatory cytokines TNF, IL-1 β and IL-6, (G) Muc5ac and (H) secretory leucocyte protease inhibitor (SLPI) were quantified by ELISA. (I) Airway hyperresponsiveness (enhanced pause [Penh]) to methacholine challenge was measured by whole body plethysmography. *n*=4-5 mice/group. Data expressed as mean \pm SEM. Data analysed by two-tailed Mann Whitney U test. **P* < 0.05, ***P* < 0.01, ****P* < 0.001

Figure 7 DNase treatment reduces immunopathology in a mouse model of virus-exacerbated COPD. (A) Experimental schematic. Mice were treated by DNaseI (or vehicle control) by i.p. injection 4 hours before inoculation and 1 and 2 days post-infection and by the intranasal route 8 hours and 1 and 2 days post-infection (B) Bronchoalveolar lavage (BAL) concentrations of dsDNA was quantified by ELISA. (C) Histological lung inflammation scores (D) BAL total cells, neutrophils, macrophages and lymphocytes were enumerated by cytopsin. BAL concentrations of (E) chemokines CXCL10/IP-10 and CCL5/RANTES and CCL2 (F) pro-inflammatory cytokines TNF, IL-1 β and IL-6 and (G) mucin glycoprotein Muc5ac and (H) secretory leucocyte protease inhibitor (SLPI) were quantified by ELISA. (I) Airway hyperresponsiveness (enhanced pause [Penh]) to methacholine challenge was measured by whole body plethysmography. n=4-5 mice/group. Data analysed by two-tailed Mann Whitney U test. * P <0.05, ** P <0.01, *** P <0.001.

C26: The authors stated that dsDNAs detected in the airways are solely-host derived, please also discuss if the dsDNAs are only released by the neutrophils or are they also likely to be released from damaged epithelial cells undergoing necrosis due to virus infection? mouse lung immunohistopathological staining/scores would somewhat provide clues to this, and the increased dsDNA observed might be due to both NETosis and damaged epithelial cells.

R26: Thank you. We have now carried out additional analyses as requested by reviewer 1 to show that when NETs are specifically blocked with a PAD4 inhibitor, this leads to almost complete suppression of dsDNA induction after viral infection, confirming that NETosis is the major source of dsDNA in this context (see new Figure 8C in revised manuscript, also copied and pasted in R6 and R7 above). In terms of cell necrosis/death, we have quantified LDH in BAL as a sensitive marker of cellular damage. These data are shown in revised Figure 4I and show reduced LDH in the elastase+virus treated groups relative to the PBS+RV group. Data are pasted below (Figure 4I).

Figure 4: Elastase treatment combined with rhinovirus infection models exacerbated neutrophils and NETs in mice. (A) Experimental schematic. Mice were treated with intranasal porcine pancreatic elastase or PBS control. 10 days later, mice were inoculated with RVA1 or UV-inactivated virus control. (B) Total neutrophil numbers in lung tissue (C) % CD63 (D) % CD64 (E) Lamp-1 and (F) CXCR4 expressing neutrophils were characterised by flow cytometry. Bronchoalveolar lavage (BAL) concentrations of (G) total intact (H3.1) (H) citrullinated (H3.R8) nucleosomes and (I) lactate dehydrogenase (LDH) were quantified by ELISA. n=4-5 mice/group. Data expressed as mean \pm SEM. Data analysed by two-tailed Mann Whitney U test. * P <0.05, ** P <0.01, *** P <0.001.

The results have been re-phrased accordingly as follows:

'Analysis of lactate dehydrogenase (LDH) concentrations in BAL found lower LDH concentrations in elastase-treated and RV infected mice versus PBS-treated RV infected control mice (Fig 4I), suggesting that the augmented dsDNA concentrations observed in RV-exacerbated mice were not due to increased cell necrosis during viral exacerbation.'

C27: However would the DNase treatment render the host unable to respond appropriately to secondary bacterial or viral infection?

R27: Thank you. DNase did not have a boosting effect upon SLPI in a similar way to the neutrophil elastase inhibitor but likewise did not inhibit this antibacterial peptide (see R21 above). To test whether DNase would promote secondary bacterial infection, an elastase-virus-bacterial coinfection model would be required. We have added this point as a future avenue within the revised discussion section.

'However, the direct implications of this effect requires formal testing using virus-bacterial co-infection mouse models and ultimately human intervention studies will be required for definitive evaluation.'

Reviewer #3 (Remarks to the Author):

C28: The authors examined the role of NETs in pulmonary inflammation using a mouse elastase model and in human samples from COPD patients.

Comments: I think the manuscript is more suitable for a specialized lung journal.

R28: We thank the author the time spent assessing our manuscript and for the comments raised, which we provide detailed responses to below. Because of this significant mechanistic insight, combined with the potential to target NETs therapeutically, we believe that the manuscript will be of broad interest to the readership of *Nature Communications*.

C29: As the authors rightly note, the role of NETs in pulmonary inflammation has already been studied in mouse models, also with rhinovirus (e.g. PMID: 28459437, 36495695). Human samples from COPD patients have also been extensively analyzed for NETs (e.g. PMID: 28506850, 25994149). The elastase model carried out in this study (which is actually more of a lung regeneration model) and the relatively small number of patient samples analyzed therefore only represents an extension of previous studies. Fig.1: Is it really surprising that COPD patients have elevated "markers" for NETs on one (day 9) of 42 days, or does this just reflect an increased number of neutrophils in the sputum of COPD patients (as shown in Fig. 6)? Why are the results actually torn apart?

R29: The references highlighted by the reviewer are all already mentioned in our manuscript and relate to measurement/roles of NETs in asthma exacerbations (PMID 28459437), stable COPD (PMID 36495695, 28506850) and a small study of exacerbated COPD of undefined aetiology in n=16 subjects (PMID 25994149). None of these existing studies specifically evaluate **virus-induced exacerbations of COPD**, which is the unique focus of our study.

We agree with the reviewer that it is not at all surprising that COPD patients have elevated markers for NETs on day 9 post-infection, given that we have previously reported increased sputum neutrophils at this time point (PMID: 25790167). We have now re-plotted the NET data (DNA-elastase complex, H3.1, and H3R8cit concentrations) adjusting for neutrophil counts (i.e on a per neutrophil basis) and show these data in supplementary figure 1 (pasted below).

We would emphasise that the important advance of the current study is not that NETs are increased in COPD exacerbations, but rather that they correlate with a range of immunological and clinical readouts and then importantly, when they are pharmacologically inhibited within a preclinical model, this directly reduces inflammatory exacerbation severity. As stated in our discussion, there has been a shift in thinking within the field away from targeting neutrophils (due to potential adverse effects upon antibacterial defence) and towards targeting NETs. We therefore would query whether the fact that increased NETs simply reflect increased neutrophils during acute exacerbation is a particularly important limitation as our study advances the field by providing rationale for targeting NETs/neutrophil products during these episodes.

This was addressed in the discussion as follows:

'Neutrophils have long been recognised to be a hallmark feature in stable COPD, present in 60-80% of patients^{4,5} and are further increased during exacerbations.^{2,6} Despite this, attempts to target neutrophilic inflammation (e.g. CXCR2 antagonists, IL-17 monoclonal antibodies) have largely failed to show efficacy in clinical trials.²³⁻²⁵ Targeting neutrophils directly may also be sub-optimal due to the delicate balance that exists between reducing inflammation versus increasing the risk of infections through immunosuppression. Therefore, focus has shifted towards downstream targets including NETs and neutrophil proteases where these risks may theoretically be lesser.'

C30: Fig.3.: Don't elevated markers for NETs simply reflect an increased number of leukocytes in the sputum? Aren't numerous inflammatory factors increased?

R30: We have plotted the data originally shown in Figure 3 (showing increased DNA-elastase complexes and citrullinated (H3.R8) nucleosomes during naturally occurring virus-induced COPD exacerbations) now corrected for neutrophil counts (note sputum neutrophil counts were only available in a subset of subjects due to variable quality of cytopins). These new data are shown in new supplementary figure 3 of the revised manuscript (pasted below) Again, there is a trend towards increases in NET markers from stable-state to exacerbation (non-significant effect, likely due to small numbers).

Supplementary Figure 3: Sputum concentrations of NETosis markers adjusted for total neutrophil counts in naturally occurring COPD exacerbations. Measurement of (A) DNA/elastase complexes (B) total intact (H3.1) and (C) citrullinated (H3.R8) nucleosomes by ELISA in sputum at stable-state, exacerbation onset and 2 weeks post-onset, divided by sputum neutrophil counts at the same timepoint.

The reviewer is also correct that numerous inflammatory factors are increased during viral exacerbations but, again, the value of our study is to dissect the relative importance of NETs/DNA in driving pathology through correlation analysis in human samples, coupled with a range of functional experiments in mouse models where we block this specific component of the inflammatory response and examine downstream readouts, reporting clear evidence of beneficial outcomes through inhibition of NET formation in virus-induced COPD exacerbation models.

C31: The number of mice with n=5 is rather small. Have the experiments been repeated?

R31: Thank you, we typically use n=5 per group with this model (see numerous examples of our prior studies in the literature e.g. PMID: 35239513, 29884817, 37857661 etc.). These numbers are based on power calculations conducted prior to commencement of the project and we always use the minimum number of animals possible to detect a significant effect, in

line with UK Home office good 3Rs ethical practice. All experiments shown in the manuscript have been repeated at least twice, as stated in the figure legends.

C32: The Elastase model is actually not a true COPD model.

R32: Thank you for this comment. We acknowledge that every model has its limitations and neither elastase (nor cigarette smoke exposure, the other main approach available) can completely and accurately recapitulate the complexities of human COPD which likely occurs in genetically susceptible individuals following prolonged (>20 years) cigarette smoke exposure. We do believe that the elastase mouse model is fit for purpose as a model for this study as it accurately reproduces immunopathological features of COPD exacerbation including histological changes, enhanced airway inflammation, mucus hypersecretion and lung function abnormalities as we have previously reported (PMID 25783022).

Our Home Office Licence does not permit use of cigarette smoke exposure and we do not have this model established in our facility so our studies to date have focused primarily on elastase-induced COPD. To obtain home office approval, establish a relevant mouse model of chronic smoke exposure (at least 3 months cumulative exposure needed) and carry out experiments would take months to years of work. We feel that this delay is unjustified as this would merely be confirming our findings in another imperfect model of COPD. We have added the following comments to the revised discussion to acknowledge limitations of the COPD mouse models:

'The elastase model of COPD, when exacerbated by RV infection, recapitulates several immunopathological features of human COPD exacerbation including augmented airway inflammation, mucus hypersecretion and lung function abnormalities. However, existing animal models including elastase or cigarette smoke exposure, are unable to fully reproduce the complex heterogenous disease features of human COPD, which occurs in genetically susceptible individuals following prolonged cigarette smoke exposure.'

C33: The fact that viruses lead to more markers for NETs in the model is not surprising given the massive neutrophilic inflammation in virus/elastase experiments. It is not clear to me how a role of NETs in pulmonary inflammation can be concluded in the model using an inhibitor of neutrophilic elastase given the massive influx of damaging elastase expressing neutrophils.

R33: We agree with the reviewer that this is not a surprising finding. Our conclusion is based on the fact that, despite the massive inflammation that is induced in virus/elastase treated mice, selective targeting of NETs using pharmacological inhibition in this model leads to widespread attenuating effects upon a range of pro-inflammatory responses/immunopathology. We would again emphasise that our conclusions in the mouse models are not based on observational data but on direct manipulation experiments to examine cause and effect. We have additionally conducted further experiments using an even more selective inhibitor of NETosis (PAD4 inhibitor), as requested by reviewer 1 and this further strengthens these conclusions.

C34: In addition, the lungs were not examined for structural changes and pulmonary function, as is usually the case in such models.

R34: Thank you. We have previously reported that the elastase protocol we adopted in this study leads to structural changes and measurable histological emphysema (reported fully in PMID 25783022) and this disease feature is fully established at the point of viral infection (10 days after elastase administration). We have also conducted prior optimisation experiments which have shown that histological changes do not progress beyond the 10 day timepoint. To determine if NET/NE inhibition alters the development of emphysema in the model, we would

have to treat mice at an earlier timepoint after elastase administration. This is a different question to the purpose of the current study which examines the effect of NET inhibition upon virus-induced immunopathology within an established chronic disease model.

With regards to pulmonary function, we have shown previously using invasive lung function testing (Flexivent) that RV infection does not exacerbate lung function (functional residual capacity, total lung capacity, dynamic compliance) in elastase treated mice (versus elastase-treated UV-RV infected controls, PMID 25783022). By contrast, airways hyper-responsiveness (AHR) assessed by whole body plethysmography is clearly exacerbated by RV infection (as reported in the same paper, PMID 25783022). We therefore deliberately chose to assess the effect of NE/DNA inhibition on RV-induced AHR by whole body plethysmography reporting clear beneficial outcomes for both NEi (Figure 5I) and DNase (Figure 7I) on this pulmonary function outcome. There would be no value to additionally assessing invasive lung function in this context, since RV infection does not lead to any augmentation of these parameters over elastase treatment alone.

REVIEWERS' COMMENTS

Reviewer #1 (Remarks to the Author):

The reviewers have appropriately addressed all my comments and were very responsive. Several new experiments and novel data have been added to the revised paper. Terms related to NETs are used in a specific manner and the text has also been rewritten when requested.

Balazs Rada

Reviewer #2 (Remarks to the Author):

The authors have responded all the comments raised and have revised the manuscript appropriately. This has led to an improved manuscript and I do not have any more questions.

Reviewer #3 (Remarks to the Author):

The authors responded extensively to my comments. The editor should decide to what extent the results presented are interesting for NC.

Response to Reviewers

REVIEWERS' COMMENTS

Reviewer #1 (Remarks to the Author):

The reviewers have appropriately addressed all my comments and were very responsive. Several new experiments and novel data have been added to the revised paper. Terms related to NETs are used in a specific manner and the text has also been rewritten when requested.

Balazs Rada

Reviewer #2 (Remarks to the Author):

The authors have responded all the comments raised and have revised the manuscript appropriately. This has led to an improved manuscript and I do not have any more questions.

Reviewer #3 (Remarks to the Author):

The authors responded extensively to my comments. The editor should decide to what extent the results presented are interesting for NC.

We thank all three reviewers for the further time spent evaluating the manuscript and for the helpful comments and suggestions which have improved the paper significantly